# Paraphrasing evades detectors of AI-generated text, but retrieval is an effective defense

**Kalpesh Krishna**♠♡∗  **Yixiao Song**♠  **Marzena Karpinska**♠
**John Wieting**◇†  **Mohit Iyyer**♠†

♠University of Massachusetts Amherst, ♡Google, ◇Google DeepMind
{kalpeshk,jwieting}@google.com
yixiaosong@umass.edu  {mkarpinska,miyyer}@cs.umass.edu

## Abstract

The rise in malicious usage of large language models, such as fake content creation and academic plagiarism, has motivated the development of approaches that identify AI-generated text, including those based on watermarking or outlier detection. However, the robustness of these detection algorithms to *paraphrases* of AI-generated text remains unclear. To stress test these detectors, we build a 11B parameter paraphrase generation model (DIPPER) that can paraphrase paragraphs, condition on surrounding context, and control lexical diversity and content reordering. Using DIPPER to paraphrase text generated by three large language models (including GPT3.5-davinci-003) successfully evades several detectors, including watermarking, GPTZero, DetectGPT, and OpenAI's text classifier. For example, DIPPER drops detection accuracy of DetectGPT from 70.3% to 4.6% (at a constant false positive rate of 1%), without appreciably modifying the input semantics.

To increase the robustness of AI-generated text detection to paraphrase attacks, we introduce a simple defense that relies on *retrieving* semantically-similar generations and must be maintained by a language model API provider. Given a candidate text, our algorithm searches a database of sequences previously generated by the API, looking for sequences that match the candidate text within a certain threshold. We empirically verify our defense using a database of 15M generations from a fine-tuned T5-XXL model and find that it can detect 80% to 97% of paraphrased generations across different settings while only classifying 1% of human-written sequences as AI-generated. We open-source our models, code and data.[1]

## 1  Introduction

Large language models (LLMs) such as ChatGPT [Schulman et al., 2022] exhibit an unprecedented ability to write coherent and relevant long-form text in response to user-specified prompts. These abilities have sparked fears of malicious applications such as automatically generating fake news articles or homework answers [Stokel-Walker, 2022]. To defend against these use cases, several algorithms have recently been proposed to detect AI-generated text, including watermarking [Kirchenbauer et al., 2023a], GPTZero [Tian, 2023], DetectGPT [Mitchell et al., 2023], and OpenAI's text classifier [OpenAI, 2023a]. However, it remains unclear how robust these algorithms are to *paraphrase attacks*, in which AI-generated text from an LLM is rewritten by another (smaller) model to convey approximately[2] the same meaning but using different word choice and syntax.

---

∗Work done as a PhD student at UMass, and partially as a student researcher in Google Research.
†John Wieting and Mohit Iyyer contributed equally as advisors.
[1]https://github.com/martiansideofthemoon/ai-detection-paraphrases
[2]We use the *quasi-paraphrase* definition of semantic equivalence [Bhagat and Hovy, 2013] in this paper.

37th Conference on Neural Information Processing Systems (NeurIPS 2023).

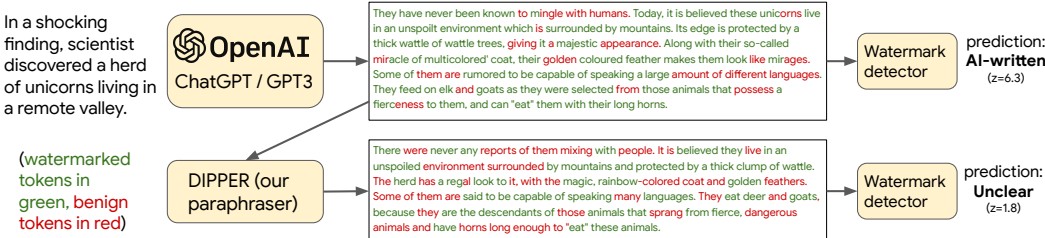

Figure 1: An overview of paraphrasing attacks with DIPPER on watermarked text [Kirchenbauer et al., 2023a]. The original model generation (top) contains several "green" watermarked tokens that are counted by a detector to judge whether the text was AI-generated. After paraphrasing, several green tokens are replaced with approximately semantically-equivalent red tokens, thereby fooling the detector. Actual outputs from a watermarked version of GPT2-XL and our paraphraser DIPPER.

In this paper, we first demonstrate the vulnerability of these existing detectors to paraphrase attacks (Section 3, 4). Such attacks require an *external* paraphraser model, since paraphrases generated by the base LLM are still susceptible to detection techniques such as watermarking. We train an 11B parameter paraphrase generation model called DIPPER (or **Di**scourse **Para**phras**er**) to execute these attacks. DIPPER possesses two unique features that help its outputs evade AI-generated text detectors: (1) **Paraphrasing long-form text in context:** Most modern paraphrasers are exclusively trained on sentence-level data, ignoring discourse-level information. However, many critical use cases of LLMs involve generating long-form text as responses to detailed user-specified prompts. Thus, we train DIPPER to paraphrase paragraph-length texts, re-order content, and optionally leverage context such as user prompts; (2) **Controlling output diversity:** Another weakness of existing paraphrasers is that they lack an easy way to control output diversity. An attacker may want to apply just the minimum amount of paraphrasing to evade a detector. DIPPER provides users with two intuitive scalar control knobs at inference time (lexical diversity, content reordering) that are trained end-to-end.

We use DIPPER to attack several recently proposed AI-generated text detectors (see Figure 1 for an attack overview). Experiments on multiple tasks and LLMs (including GPT3.5-davinci-003) show that after paraphrasing with DIPPER, a substantial fraction of AI-generated texts are misclassified as human-written texts by all detectors. For example, DetectGPT [Mitchell et al., 2023] correctly detects 70.3% of AI-generated sequences from GPT2-XL, but after paraphrasing, its detection rate drops to only 4.6%[3] despite minimal semantic modification. We confirm the validity of DIPPER's paraphrases through several automatic evaluations and a human evaluation of semantic similarity.

Given the vulnerability of AI-generated text detectors to paraphrasing, how can we defend against such attacks? In the second part of our paper (Section 5), we propose to use *retrieval* methods to detect AI-generated text instead of relying on statistical properties of the text or watermarking. First, an LLM API provider stores every output generated by their model in a database. The API provider then offers a service in which a semantic representation of a candidate text is compared to representations of every generation stored in the database. The search focuses on the *semantics* of the input and can leverage both standard IR methods such as BM-25 [Robertson et al., 1995] as well as semantic vector representations such as P-SP from Wieting et al. [2022]. Since paraphrasing does not modify the semantics of the input, this algorithm is robust to paraphrasing attacks. Specifically, we find that 97.3% of PG19 paraphrases and 80.4% of Wikipedia paraphrases are successfully detected in a large database of over 15M generations, at a 1.0% false positive rate. We extensively discuss the limitations and scalability of retrieval-based detection in Section 5.4.

In contrast to concurrent work that also uses paraphrasing to attack AI-generated text detectors [Sadasivan et al., 2023], our work offers more comprehensive attack experiments, a new and more powerful paraphraser, human evaluations of paraphrase quality, and finally a novel defense mechanism based on retrieval to combat such attacks. To spur future research in this area, we will release our DIPPER model, data, and a codebase for evaluating both existing detectors and our retrieval-based method.

---

[3]These detection rates were computed at a constant false positive rate (FPR) of 1%. Due to the importance of low FPR in this task, we recommend using a fixed low FPR rather than AUC-ROC values; see Section 4.1.

## 2 Background on detectors of AI-generated text

In this section, we provide a brief overview of existing algorithms for detecting AI-generated text detection (see Appendix E for a detailed version). We also contrast our work to Sadasivan et al. [2023], a concurrent effort which notes the efficacy of paraphrasing attacks but does not consider a retrieval-based defense in its pessimistic conclusion about the fate of AI-generated text detection.

A **watermark** is a modification to the generated text that can be detected post-hoc by an algorithm while remaining imperceptible to human readers. Effective watermarks are difficult to remove and have little effect on the quality of generated text. Prior work has watermarked natural language using syntax tree manipulations [Topkara et al., 2005, Meral et al., 2009], and this area has received renewed interest with the advent of LLMs [Abdelnabi and Fritz, 2021, Grinbaum and Adomaitis, 2022]. Most recently, Kirchenbauer et al. [2023a] proposed a simple algorithm that watermarks LLMs by slightly perturbing its probability distribution while generating text.

**Statistical outlier detection methods** detect AI-generated text based on its artifacts [See et al., 2019, Holtzman et al., 2020] instead of modifying the generative algorithm. Early methods detect statistical irregularities in entropy [Lavergne et al., 2008] and perplexity [Beresneva, 2016], while Gehrmann et al. [2019] introduced the GLTR visualizer to assist humans in detecting AI-generated text. The release of ChatGPT prompted the development of two new tools: closed-source GPTZero [Tian, 2023] and open-source DetectGPT [Mitchell et al., 2023]. The latter uses the observation that AI-generated text tends to have significantly higher LLM likelihood than meaningful perturbations of it.

**Classifier methods** train models to distinguish human-written text from AI-generated text. Early efforts used classifiers to detect fake reviews [Hovy, 2016] and news [Zellers et al., 2019], while others examined classifier performance across domains [Bakhtin et al., 2019] and decoding strategies [Ippolito et al., 2020]. Most recently, OpenAI fine-tuned a GPT model to perform this task and released it as a web interface [OpenAI, 2023a]. Their model uses generations from 34 LLMs, with the human-written text from Wikipedia, WebText, and their internal human demonstration data.

**Comparison to Sadasivan et al. (2023)**: In recent concurrent work, Sadasivan et al. [2023] also demonstrate the utility of paraphrasing attacks against AI-generated text detectors. Our experiments encompass more tasks, detection algorithms, and larger LMs like GPT3.5. Additionally, we propose a discourse-level paraphrase model (DIPPER) that is much more suited to long-form text than the off-the-shelf sentence-level paraphrasers used in their paper. More importantly, our retrieval-based defense *directly contradicts* the "impossibility result" of Sadasivan et al. [2023] and its associated proof, which states that an optimal detector will perform at random as the quality of LLM-generated text approaches that of human-written text. The quality of generated text is irrelevant to our detector's accuracy because it relies only on a corpus search, and thus the proof is inapplicable. Other concurrent work [Chakraborty et al., 2023] has also shown the proof's invalidity in practical settings.

## 3 Building a controllable discourse paraphraser

Having outlined existing methods to detect AI-generated text, we now focus on a simple attack against all detection techniques: *paraphrasing* the generated text. Intuitively, paraphrasing alters the statistical properties of AI-generated text, which can fool outlier detection or classifiers while also reducing the number of watermarked tokens (Figure 1). To evade such detectors, a paraphraser must be able to handle *context* in the form of prompts or multi-sentence inputs. Its behavior should also be *controllable* in order to make as many/few changes as needed to evade a given detector. In all cases, it should not appreciably change the input semantics. Finally, to evade watermarking, the paraphraser must be different from the watermarked model, as otherwise the paraphrases will also be watermarked. Below, we detail how we construct a paraphraser (DIPPER) with all these properties.[4]

**Constructing paraphrase data**: Our process involves fine-tuning a LLM on a parallel dataset of paragraph-level paraphrases, which we modify to model control, external context and content reordering. We leverage the PAR3 dataset [Thai et al., 2022], which contains multiple translations of non-English novels into English aligned at a paragraph level, which we treat as paraphrases. More formally, let $p$ and $q$ be aligned paragraphs, where $p_1, p_2...p_N$ denote sentences of $p$ and $q_1, q_2, ...q_M$

---

[4]To better ground DIPPER's abilities in prior work, we survey existing paraphrase models in Appendix D.1.

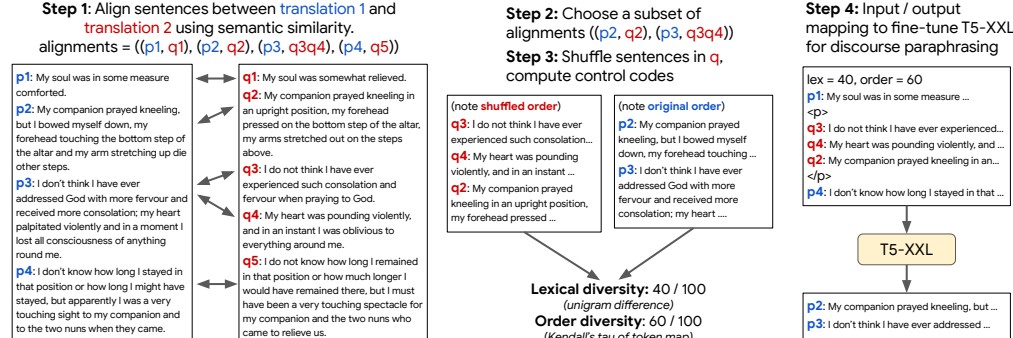

Figure 2: The method used to train DIPPER on English translations of the French novel *The Nun*. We first align sentences between the two translations to create parallel data. Next, a subset of the alignments are chosen; in this example, we use $(p_2, q_2)$ and $(p_3, q_3q_4)$. We shuffle sentences, compute control codes, and fine-tune a T5-XXL LM to generate $p_2p_3$ given $q_3q_4q_2$ and the context $p_1$ and $p_4$.

denote sentences of $q$. Note that $M$ may not be equal $N$ when two translators disagree on when to merge and split sentences. We perform the following steps (overview in Figure 2):

1. **Align sentences** of $p$ to sentences of $q$ by using the semantic similarity scores from the paraphrase similarity metric in Wieting et al. [2019] to run the sequence alignment algorithm designed by Needleman and Wunsch [1970] which uses dynamic programming (metric details in Section 4.1).

2. **Choose a subset of sentences** $p_i...p_j$ from the first paragraph. Let $q_{i'}...q_{j'}$ be the corresponding alignment in the second paragraph. In Figure 2, $i = 2, j = 3, i' = 2, j' = 4$.

3. **Re-order** the sentences $q_{i'}...q_{j'}$ randomly, and compute the **diversity control codes** between $p_i...p_j$ and shuffle$(q_{i'}...q_{j'})$. We shuffle the sentences to allow for the model to learn content re-ordering. We compute lexical diversity ($L$) using unigram token overlap (F1 score), and the order diversity ($O$) using the Kendall-Tau correlation of tokens of overlapping words between $p_i...p_j$ and shuffle$(q_{i'}...q_{j'})$, also used in Krishna et al. [2020]. These scores are normalized to values $\{0, 20, 40, 60, 80, 100\}$, where $L = 20$ roughly corresponds to a 20% lexical modification.

4. **Map** the shuffled $q_{i'}...q_{j'}$ to $p_i...p_j$, leveraging context from the rest of $p$ and control codes using string concatenation. Let input = shuffle$(q_{i'}...q_{j'})$. We map,

$$\text{lexical} = L, \text{order} = O \oplus p_1...p_{i-1} \oplus \texttt{<p>} \text{ input } \texttt{</p>} \oplus p_{j+1}...p_N \longrightarrow p_i...p_j$$

where $\oplus$ is string concatenation. During inference, we can paraphrase any sequence of sentences by marking it with `<p>` tags, assigning the control codes ($L$, $O$) the desired diversity values.

Our final dataset contains 6.3M paraphrase pairs. We **fine-tune** a sequence-to-sequence Transformer [Vaswani et al., 2017] on this data, initialized with the pretrained 11B parameter T5-XXL checkpoint [Raffel et al., 2020]. See Appendix F.1 for details.

## 4 Experiments attacking detection algorithms with DIPPER

In this section, we describe our experimental setup in Section 4.1-4.2 and present our results in Section 4.3. Overall, we find that DIPPER evades all detectors across three LLMs (including GPT3.5).

### 4.1 Evaluation metrics

**Detection accuracy**: Our first metric measures how often the input text is correctly detected as AI-generated. Since detection rates are heavily dependent on the chosen detection threshold, the AUC-ROC metric is commonly used to measure detector performance [Mitchell et al., 2023], which considers the range of all possible thresholds. However, in this application, it is critical that the *false positive rate* (FPR) is low; in other words, human-written text must almost never be classified as machine-generated [OpenAI, 2023a, Kirchenbauer et al., 2023a]. Hence, we fix the FPR to 1% for all detection algorithms (although even 1% is likely too high in practice), and adjust the detection threshold accordingly while reporting detection accuracies. Additionally, we also plot ROC curves focusing on the 0-1% FPR region. Overall, we expect detection rates to plummet on paraphrased text.

Table 1: Performance of detection algorithms (at 1% FPR) before and after DIPPER paraphrasing on **open-ended generation** using Wikipedia prompts (300 generated tokens). As the diversity (L,O) increases, detection rates decrease across algorithms, with nearly perfect semantic similarity (Sim). *GPT3.5 DetectGPT scores computed using 200 samples at 20% FPR as it scores 0% at a 1% FPR.

| Metric → | Sim ↑ | Detection Accuracy ↓ | | | | |
|---|---|---|---|---|---|---|
| Detector → | | Watermarks | DetectGPT | OpenAI | GPTZero | RankGen |
| GPT2-1.5B | - | 100.0 | 70.3 | 21.6 | 13.9 | **13.5** |
| + DIPPER 20L | 99.2 | 97.1 | 28.7 | 19.2 | 9.1 | 15.8 |
| + DIPPER 40L | 98.4 | 85.8 | 15.4 | 17.8 | 7.3 | 18.0 |
| + DIPPER 60L | 96.9 | 68.9 | 8.7 | **13.3** | 7.1 | 19.8 |
| + DIPPER 60L, 60O | 94.3 | **57.2** | **4.6** | 14.8 | **1.2** | 28.5 |
| OPT-13B | - | 99.9 | 14.3 | 11.3 | 8.7 | **3.2** |
| + DIPPER 20L | 99.1 | 96.2 | 3.3 | 11.8 | 5.4 | 5.2 |
| + DIPPER 40L | 98.6 | 84.8 | 1.2 | 11.6 | 3.8 | 6.6 |
| + DIPPER 60L | 97.1 | 63.7 | 0.8 | **9.1** | 6.3 | 9.3 |
| + DIPPER 60L, 60O | 94.6 | **52.8** | **0.3** | 10.0 | **1.0** | 13.5 |
| GPT-3.5-175B, davinci-003 | - | - | 26.5* | 30.0 | 7.1 | **1.2** |
| + DIPPER 20L | 97.6 | - | 12.5* | 20.6 | 4.3 | 1.7 |
| + DIPPER 40L | 96.7 | - | 8.0* | 22.4 | 4.8 | 2.0 |
| + DIPPER 60L | 94.2 | - | 7.0* | **15.6** | 6.1 | 3.9 |
| + DIPPER 60L, 60O | 88.4 | - | **4.5*** | **15.6** | 1.8 | 7.3 |
| Human Text | - | 1.0 | 1.0 | 1.0 | 1.0 | 1.0 |

**Semantic similarity (Sim)**: Detection accuracy is an insufficient evaluation of our attack's success. We also need to measure whether the original and paraphrased generations share approximately the same semantics. We measure semantic similarity using the state-of-the-art semantic similarity model P-SP from Wieting et al. [2022], an embedding averaging model trained on a large corpus of filtered paraphrase data [Wieting and Gimpel, 2018]. P-SP is a well-calibrated metric that performs well on semantic calibration tests as well as plagiarism detection in STS benchmarks [Agirre et al., 2016]. P-SP is also robust against topically similar non-paraphrases. We found that P-SP it scores just 0.09 on random pairs of paragraphs from the same book (topically similar paragraphs but not paraphrases) in the PAR3 dataset [Thai et al., 2022]. In contrast, the average P-SP score of actual human paraphrase pairs in PAR3 is 0.76. We consider semantics to be approximately preserved if the P-SP score is greater than this average human paraphrase score of 0.76.

Besides semantic similarity, we conduct several automatic evaluations, ablation studies, and human evaluations of intrinsic paraphrase quality in Appendix C.

### 4.2 Models, datasets & detection algorithms

**Base language models**: We conduct attacks on three language models of varying sizes that belong to different model families. We consider the GPT2-XL model (1.5B parameters) [Radford et al., 2019], the OPT-13B model [Zhang et al., 2022], and the `text-davinci-003` variant from the GPT-3.5 family [Brown et al., 2020], which has 175B parameters and has additionally been instruction tuned using reinforcement learning from human feedback [Ouyang et al., 2022]. For all LMs, we sample generations that are 300 tokens long before passing them through DIPPER for the attack experiments.[5]

**Natural language generation tasks**: We experiment with two long-form text generation tasks, since most malicious applications (e.g., fake article creation) are associated with long-form outputs. First, we consider *open-ended generation*, where an LM generates a continuation to a two-sentence prompt. To obtain our prompts, we sample 3K contiguous two-sentence chunks from the validation split of WikiText-103 [Merity et al., 2017] and use the next 300 tokens as the "human-written" continuation. Second, we evaluate *long-form question answering* [Fan et al., 2019], in which an LM answers a question with a 300-word answer (dataset details in Appendix F.2). For our main results, the human

---

[5]For GPT2-XL and OPT-13B, we generate text using nucleus sampling [Holtzman et al., 2020] with $p = 0.9$. For `davinci-003`, we use the default hyperparameters on the API Playground (temperature = 0.7).

reference answers or continuations are only used to adjust detection thresholds of studied methods to maintain a 1% FPR.[6] Note that we are not removing human-written text from our test set. Our metric is equivalent to having a test set with a 50-50 split between machine/human-written text for the same prompts, and observing the FPR=1% point in the ROC curve (also provided in Appendix H).

**Detection algorithms**: We attack five detectors:[7] (1) watermarking [Kirchenbauer et al., 2023a]; (2) DetectGPT [Mitchell et al., 2023]; (3) GPTZero [Tian, 2023]; (4) OpenAI's text classifier [OpenAI, 2023a];[8] and (5) RankGen-XL-all [Krishna et al., 2022a].[9] We use the default hyperparameters for each detector. We also respect their minimum length specifications, discarding instances where any of the AI-generated text, human-written text, or paraphrased text is shorter than the minimum length.

**Paraphrasing AI-generated text**: We pass the prompts for each task and AI-generated responses to those prompts through DIPPER. Specifically, we feed the model input of the form,

> lexical = L, order = O prompt
> <p> generated-text </p>,

where $L$ and $O$ represent the paraphraser diversity control codes and the <p> and </p> special tokens mark the boundaries of the text to be paraphrased. We use $L = 20, 40, 60$ and $O = 0, 60$ in our main attack experiments. After paraphrasing, we ensure that the AI-generated sequence, paraphrased sequence, and human-written sequence have an equal number of words by truncating them to the length of the shortest among the three. To ensure higher semantic preservation, we iteratively paraphrase long sequences three sentences at a time, keeping already paraphrased text in the context of the generation. To show the effectiveness of our attack, we only **paraphrase each generation once**, rather than draw multiple samples until it evades detection.[10]

Figure 3: Detector performance (at 1% FPR) on **long-form QA** before/after paraphrasing. As diversity (L,O) increases, detection rates decrease with very high semantic preservation (Sim). WM: Watermark, D.GPT: DetectGPT, O.AI: OpenAI. *GPT3.5 D.GPT uses 100 samples at 20% FPR to show attack success, as it scores 0% at 1% FPR.

| Metric → | Sim ↑ | Detection Accuracy ↓ | | |
|---|---|---|---|---|
| | | W.M. | D.GPT | O.AI |
| GPT2-1.5B | - | 100.0 | 74.9 | 59.2 |
| + DIPPER 20L | 99.5 | 98.9 | 45.7 | 35.3 |
| + DIPPER 40L | 99.0 | 90.7 | 28.0 | 34.4 |
| + DIPPER 60L | 97.5 | 71.1 | 15.8 | **31.3** |
| + 60L, 60O | 96.2 | **55.8** | **7.6** | 32.7 |
| OPT-13B | - | 100.0 | 29.8 | 33.5 |
| + DIPPER 20L | 99.6 | 98.3 | 15.0 | 24.5 |
| + DIPPER 40L | 99.4 | 87.3 | 6.4 | 24.1 |
| + DIPPER 60L | 96.5 | 65.5 | 3.2 | **21.6** |
| + 60L, 60O | 92.9 | **51.4** | **1.5** | **21.6** |
| GPT-3.5-175B davinci-003 | - | - | 67.0* | 40.5 |
| + DIPPER 20L | 99.9 | - | 54.0* | 43.1 |
| + DIPPER 40L | 99.8 | - | 36.0* | 43.1 |
| + DIPPER 60L | 99.5 | - | 23.0* | 40.1 |
| + 60L, 60O | 98.3 | - | **14.0*** | **38.1** |
| Human Text | - | 1.0 | 1.0 | 1.0 |

## 4.3 Attacking AI-generated text detectors

We present our results in Table 1 and Figure 3. Overall we find that:

**Paraphrasing significantly lowers detection accuracy while preserving input semantics**. Across all LMs, detectors,[11] and tasks, paraphrasing significantly lowers detection accuracy across all diversity control codes. For instance, paraphrasing GPT2-XL open-ended generations reduces watermark detection accuracy from 100% to 57.2%, and DetectGPT accuracy from 70.3% to just 4.6%. Trends are similar even for large LMs like GPT-3.5, for which paraphrasing reduces OpenAI's text classifier accuracy from 30.0% to 15.6%. Additionally, DIPPER preserves semantics effectively,

---

[6]Alternatively, if a random half of the human-written data was used for threshold adjustment, we find the other half has a FPR of 0.8%-1.2% across random splits, and this deviation will reduce with a bigger dataset.

[7]We consider both model-specific and model-agnostic detectors, as justified in Appendix F.3.

[8]This classifier was taken down in July 2023 by OpenAI due to its low accuracy.

[9]While RankGen was not explicitly optimized for this task, it was trained to identify well-written continuations, so we hypothesize that it could also act as a reasonable AI-generated text detector.

[10]We discuss this attack briefly in Section 4.4.

[11]Except RankGen, which scores paraphrases as AI-generated more often than non-paraphrased text. We attribute this to paraphrases being poorer continuations to the prompt compared to the original (Appendix C), an aspect RankGen bases its score on. However, it has low overall performance since it is not trained for this task.

as 88%-99% paraphrases achieve a P-SP score higher than the median score of human-written paraphrases. High semantic preservation is supported by careful human evaluations in Appendix C.2. Overall, we find that watermarking is the most resilient detector to paraphrasing.

**Non-watermarking detectors are generally ineffective.** We observe that all detectors apart from watermarking struggle with text generated by larger models like OPT-13B and GPT-3.5, achieving detection accuracies < 50%. While DetectGPT is effective on the smaller GPT2-XL model (74.9% on long-form QA), its accuracy drops to just 29.8% on OPT-13B. Furthermore, GPTZero and RankGen perform the worst among the five detectors on all tested LMs (Table 1), as they are only able to detect < 15% of non-paraphrased AI-generated text. Thus, we recommend against using these detectors.

**ROC plots confirm the trends at different false positive rates**. In Figure 4, we plot the detection accuracy (true positive rate) at different values of FPR between 0% and 1% for GPT2-XL. Overall, paraphrasing significantly drops detection rates across all FPR thresholds (more plots in Appendix H).

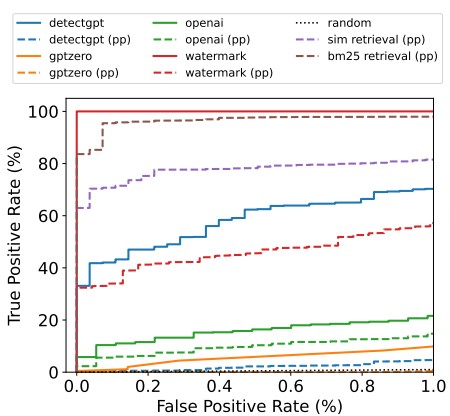

Figure 4: ROC plots (0-1% FPR) for GPT2-XL using different detectors, before (solid lines) and after paraphrasing (dashed). Paraphrasing reduces detection rate across FPRs, and our detector *retrieval* detects paraphrases best; full plots in Appendix H.

### 4.4 Alternative paraphrasing attacks

**Paraphrasing multiple times:** Our presented attacks use just a single paraphrase generated by DIPPER to evade detection. A simple way to further improve the effectiveness of a paraphrase attack is to sample multiple times[12] from DIPPER and choose a paraphrase that evades the detector while also preserving semantics. We do not perform this attack as it can only be done if an attacker has access to a detector, which may be a strong assumption (see Appendix A.2). That being said, using multiple paraphrase samples can make the attacks even more potent against publicly available detectors.

**Non-DIPPER paraphrasers:** A second alternative is to use non-DIPPER paraphrasers that operate at the sentence level. These models can be deployed for long-form text inputs by paraphrasing the inputs sentence by sentence, ignoring prompt context. While the concurrent work of Sadasivan et al. [2023] shows that this method can also evade detection, our ablations in Appendix C show that non-contextual versions of DIPPER have lower quality and are less compatible with the prompt as DIPPER paraphrasers. Moreover, most existing paraphrasers lack fine-grained diversity control and multi-sentence input support (survey in Appendix D.1), two desired properties from an attacker's point of view: attackers want to modify long multi-sentence responses *just enough* to evade detection.

A more interesting option is to use an LLM like ChatGPT to perform few-shot contextual paraphrasing. While this method is likely to provide accurate paraphrases,[13] they may be detectable by strategies like watermarking (whether using the same API as the original LLM or a different one). We thus expect a sophisticated adversary to use their own private paraphraser (like DIPPER) to evade detection.

## 5 Defense against paraphrase attacks using retrieval

In Section 4.3, we showed that paraphrasing is an effective attack against AI-generated text detectors. How can LLM API providers defend against these attacks? In this section, we propose *retrieval* over previously-generated sequences as a defense against paraphrase attacks. At a high level (Figure 5), an API provider first stores every sequence generated by their LLM in a database. The API provider offers an interface that allows users to enter candidate AI-generated text as a query. The interface searches

---

[12]Precisely, compute $f_{\text{dipper}}(x)$ for different random seeds while sampling text. Alternatively, an attacker could also compute $f_{\text{dipper}}(f_{\text{dipper}}(...f_{\text{dipper}}(x)))$, but this will lead to excessive semantic drift from $x$.

[13]In initial experiments, we observed that DIPPER performs competitively with the much larger and more powerful GPT-3.5 davinci-003 model in terms of paraphrase quality, and significantly better at controlling diversity. This finding shows that specialized smaller models can outperform LLMs in paraphrasing tasks.

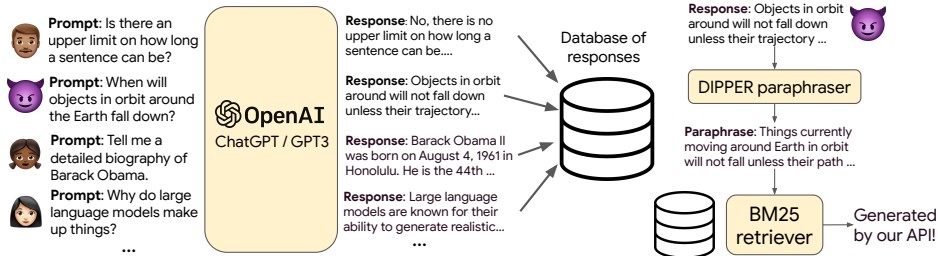

Figure 5: An illustration of AI-generated text detection with retrieval. Several users (including the attacker, shown as the purple emoji) feed prompts to the API which are collectively added to a private API-side database. Candidate queries are compared against this database using a retriever like BM25.

over the entire database of previously-generated text, trying to find a sequence that approximately matches the content of the input query. This search can be done using a semantic similarity scorer like P-SP [Wieting et al., 2022] or a retriever like BM25 [Robertson et al., 1995]. Since paraphrasing approximately preserves input semantics, we expect such a defense to still be able to map paraphrased generations to their source. We formalize our detector in Section 5.1, and then conduct a controlled comparison with competing detectors in Section 5.2. We evaluate retrieval-based detection at scale using a large retrieval corpus of 15M generations in Section 5.3. In Appendix A we extensively discuss limitations of retrieval-based detection and share ideas for enabling further scaling.

## 5.1 Formulating the retrieval defense

Let $f_{\text{LM}}$ be an LLM API (e.g., GPT-3.5) that takes a prompt $x$ as input and returns a continuation $y$. Let $f_{\text{ret}}$ be an encoder (e.g., TF-IDF, neural network) that embeds variable-length sequences into fixed-size vectors that represent the input semantics. Then, we do the following:

**Building the database**: Let $x_1, ..., x_N$ be the set of prompts that have been fed as input to the API in the past with $y_i = f_{\text{LM}}(x_i)$ being the LLM output. Here $N$ can potentially be very large for popular APIs (we study up to $N = 15\text{M}$). We construct our database $\mathbf{Y} = [\mathbf{y}_1, ...\mathbf{y}_N]$ by encoding every LLM API output with our retrieval encoder, or $\mathbf{y}_i = f_{\text{ret}}(y_i)$. The database $\mathbf{Y}$ is dynamically updated and stored on the API side. It is inaccessible to clients except via the API described in the next step.

**Querying the database**: Let $y'$ be a candidate text and $\mathbf{y}' = f_{\text{ret}}(y')$ its encoded vector. Suppose a client wishes to know whether $y'$ was generated by the API $f_{\text{LM}}$. The API provider can check this by seeing whether the maximum similarity score of $y'$ to an entry in the database exceeds some detection threshold $T$ chosen by the API provider:

$$\text{output} = \text{score} > T, \text{ where score} = \max_{i \in \{1,..N\}} \frac{\mathbf{y}' \cdot \mathbf{y}_i}{|\mathbf{y}'| \, |\mathbf{y}_i|}$$

We expect unperturbed machine-generated text to always get a score of 1.0, while paraphrasing the text may lower the detection score. Hence, lowering $T$ will increase the detection rate of heavily-paraphrased text but also increase the false positive rate (i.e., human-written text that resembles sequences previously generated by the LLM API can be falsely flagged). Since $N$ can be very large, the score can also be approximated using efficient nearest neighbor libraries like FAISS [Johnson et al., 2019]. However, in this work we only compute exact inner products.

As the **retriever** $f_{\text{ret}}$, we experiment with two choices: P-SP [Wieting et al., 2022] and BM25 [Robertson et al., 1995]. We implement BM25 using the `retriv` library from Bassani [2022]. In order to normalize and calibrate BM25 scores, we compute the F1-score unigram token overlap [Rajpurkar et al., 2016] between the candidate $y'$ and the best retrieval $y*$ to get a detection score in $[0, 1]$.

## 5.2 Controlled comparisons of retrieval with other AI-generated text detectors

First, we conduct a controlled comparison between the detection algorithms evaluated in Section 4.3 and our retrieval method on long-form question answering.[14] We construct three kinds of databases,

1. 3K sequences generated by a specific LM for one of the tasks;

---

[14]Corresponding results in open-ended text generation on Wikipedia prompts are provided in Appendix G.1.

Table 2: A comparison of retrieval against other detectors on long-form QA (300 generated tokens). Our detector outperforms baselines (at 1% FPR) even with the most diverse paraphrases (+60L,O).

| | GPT2-XL | | | OPT-13B | | | GPT-3.5 (davinci-003) | | |
|---|---|---|---|---|---|---|---|---|---|
| | Original | + 60L | + 60 L,O | Original | + 60L | + 60 L,O | Original | + 60L | + 60 L,O |
| Watermark [2023a] | 100.0 | 71.1 | 55.8 | 100.0 | 65.5 | 51.4 | - | - | - |
| DetectGPT [2023] | 74.9 | 15.8 | 7.6 | 29.8 | 3.2 | 1.5 | 1.0 | 0.0 | 0.0 |
| OpenAI [2023a] | 59.2 | 31.3 | 32.7 | 33.5 | 21.6 | 21.6 | 40.5 | 40.1 | 38.1 |
| *(Ours)* Retrieval over corpus of 3K generations from model itself, with retriever: | | | | | | | | | |
| SP | 100.0 | 95.6 | 87.7 | 100.0 | 94.8 | 85.3 | 100.0 | 94.2 | 85.1 |
| BM25 | 100.0 | 99.2 | 97.8 | 100.0 | 99.3 | 97.3 | 100.0 | 98.6 | 96.2 |
| *(Ours)* Retrieval over corpus of 9K generations pooled from all three models, with retriever: | | | | | | | | | |
| SP | 100.0 | 88.9 | 75.4 | 100.0 | 89.6 | 76.4 | 100.0 | 93.8 | 84.6 |
| BM25 | 100.0 | 98.3 | 95.2 | 100.0 | 98.5 | 94.4 | 100.0 | 98.5 | 96.0 |
| *(Ours)* Retrieval over 43K ShareGPT responses + corpus of 3K generations from model itself, with retriever: | | | | | | | | | |
| SP | 100.0 | 94.0 | 84.8 | 100.0 | 94.2 | 84.7 | 100.0 | 94.1 | 84.9 |
| BM25 | 100.0 | 98.9 | 97.5 | 100.0 | 99.0 | 97.3 | 100.0 | 98.4 | 95.5 |

2. 9K sequences formed by concatenating the generations from all three LMs in this paper;

3. 46K sequences constructed by combining the 3K sequences from (1) with 43K LLM responses from ShareGPT-Vicuna.[15]

We expect (2) to be a more difficult test for our method than (1), since the retriever needs to distinguish between multiple generations from different models given the same prompt. On the other hand, (3) denotes a more real-world setting with several diverse LLM-generated outputs from ShareGPT. Next, we perform retrieval over this corpus using different types of queries: the original AI-generated text, its DIPPER paraphrase, and human-written text (each query with at least 50 tokens).

Table 2 shows that **across all LMs, retrieval is a much more effective detector than baseline detectors**. On unperturbed AI-generated text, retrieval has a 100% detection accuracy due to exact match with the retrieval corpus. On paraphrased text, retrieval with BM25 is quite effective, detecting 97.8% of the highest-diversity paraphrases (L60, O60) on GPT2-XL, 97.3% on OPT-13B and 96.2% on GPT-3.5 in long-form question answering. This is significantly better than the next best alternative with competing detectors (55.8%, 51.4%, 38.1%). Even on our harder augmented databases, detection rates continue to be high: 95.2%, 94.4%, 96.0% for the 9K augmented database; 97.5%, 97.3%, 95.5% for the ShareGPT augmented database. Finally, we observe that BM25 is a more effective retriever than P-SP, scoring 95.2% vs 75.4% on the augmented setting in GPT2-XL. These trends are consistent across different FPR thresholds, as shown in Figure 4.

In Appendix G.2, we additionally observe promising preliminary results that show the effectiveness of retrieval against **text mixing attacks** [Kirchenbauer et al., 2023b].

### 5.3 Is retrieval an effective detector with a large retrieval corpus?

In the previous section, we conducted experiments using the set of 9K sequences generated by all three models as the retrieval corpus. However, this is more of a toy experiment: in practice, a popular LLM API may serve millions of queries a day. As the corpus grows larger, the false positive rate (i.e., human-written text falsely detected as AI-generated) will grow. How well do retrieval-based detectors scale? To answer this question, we need access to a large corpus of AI-generated text. We utilize the training data used to train RankGen [Krishna et al., 2022a], which contains over 70M AI-generated sequences. We use the Project Gutenberg and Wikipedia splits of the training data, each of which contain 15M sequences generated by a T5-XXL model [Raffel et al., 2020] fine-tuned on the different documents in the same domain. We discard generations which are shorter than 50 tokens, and paraphrase a subset of 2K generations to evaluate retrieval.

**Retrieval is effective even with a corpus size of 15M generations.** In Figure 6a, we plot the detection accuracy as a function of retrieval database size. Overall, we observe that detection

---

[15] huggingface.co/datasets/anon8231489123/ShareGPT_Vicuna_unfiltered

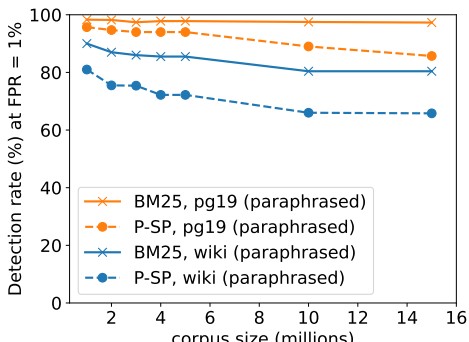
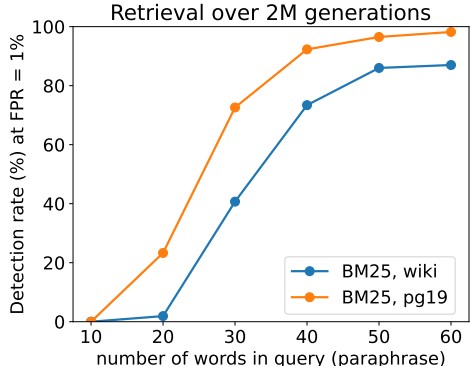

(a) Variation in retrieval-based detection with retrieval corpus size. Note consistently high detection rates for paraphrases, which only slightly degrades as the corpus is scaled to 15M generations.

(b) Variation in retrieval-based detection with different query lengths. Overall, retrieval performs best with queries of length 50+ tokens.

Figure 6: Detection rate using retrieval at 1% FPR w.r.t. corpus size (left) and query length (right).

accuracy remains consistently high across different corpus sizes (varying from 1M generations to 15M generations). We observe slight drops in performance as the corpus size increases: just 1% (98.3 to 97.3) on Project Gutenberg (PG19) and 9.6% (90.0 to 80.4) on Wikipedia. Consistent with the results in Section 5.2, BM25 continues to outperform P-SP across different corpus sizes.

**Retrieval detection works best with 50 or more tokens of generated text**. Another important factor for our retrieval-based detector is the query length: shorter queries are likely to have more matches (many of them spurious) compared to longer ones. In Figure 6b, we plot the detection accuracy of paraphrased sequences at various query lengths by truncating each sequence to its first $X$ words before using it as a query for BM25. We use a retrieval corpus of 2M generations for this experiment. We observe that BM25 struggles to detect paraphrased text with a query length of 20 (less than 25% accuracy), but the detection rate rapidly increases and begins to plateau at 50 tokens.

## 5.4 Scalability and limitations of retrieval-based detectors

In Appendix A we extensively discuss the scalability of retrieval (A.1), its limitations (A.2), ideas for improving retrieval-based detectors (A.3), and incentive structures for LLM providers to implement retrieval (A.4). In summary, we believe retrieval-based detection is a scalable approach: we estimate that if OpenAI implemented it with ChatGPT, they would need just 5TB of storage space per month (similar to modern portable hard disks). Furthermore, retrieval on ChatGPT scale takes 130 seconds per retrieval on a CPU-only Macbook Pro, which can certainly be further optimized. However, retrieval-based detection has some important limitations: (1) potential privacy risk of exposing *all* LLM responses behind a binary classifier; (2) inability to use retrieval-based detection on open-source LLMs like LLAMA [Touvron et al., 2023]; and (3) the need to implement and maintain retrieval infrastructure. We discuss mitigation strategies for the limitations in Appendix A.2.

## 6 Conclusion

We present DIPPER, a controllable paraphraser that can rewrite paragraphs in context. We use DIPPER to stress test current AI-generated text detectors, and we find that DIPPER paraphrases easily evade these detectors while preserving input semantics. As a defense, we propose a simple retrieval-based detector which searches through a corpus of previously-generated sequences from an LLM API for semantically-similar generations to a given query. We show that this defense significantly outperforms baselines on paraphrased text, and scales effectively. We discuss the limitations and ethical considerations of our work in Appendix A, B. We have additionally open sourced our models, code and data to enable future research.[1] Since this paper's initial release, DIPPER has been extensively utilized in follow-up studies to measure the robustness of AI-generated text detection algorithms ([Lu et al., 2023, Koike et al., 2023, Zhao et al., 2023, Yoo et al., 2023, Patil et al., 2023, Kirchenbauer et al., 2023b, Kumarage et al., 2023, Liu et al., 2023], to name a few).

## Acknowledgements

We are very grateful to our anonymous reviewers for their detailed feedback which helped improve our paper. We also thank Kenton Lee, Katherine Thai, Slav Petrov, Fernando Pereira, Jon Clark, William Cohen, Tu Vu, Yapei Chang, Shufan Wang and the UMass NLP group for several useful discussions on earlier versions of this paper. Kalpesh Krishna was supported by a Google PhD Fellowship. Part of this work was done while Kalpesh was a Student Researcher at Google Research, hosted by John Wieting.

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

# Appendix

## A   Limitations of retrieval-based detection and ideas for scaling it further

This section extensively discusses the scalability, limitations, future work and and an LLM API provider's incentive structures for retrieval-based detection. First, in Appendix A.1, we discuss the scalability of retrieval-based detection in terms of compute requirements, storage space, and accuracy on larger databases. Next, in Appendix A.2, we first point out some limitations of using retrieval for AI-generated text detection (Section 5), some of which potentially apply to all existing detectors. Along with limitations, we provide several possible workarounds. In Appendix A.3, we then discuss ideas that can make the proposed retrieval detection work well at an even larger scale than the one we discussed in Section 5. Finally, in Appendix A.4 we briefly discuss why an LLM provider may be incentivized to implement retrieval-based detection, and the relationship of this detector with GDPR's right to be forgotten.

### A.1   Scalability of retrieval-based detection

Retrieval-based detection requires the storage of a large database of LLM-generated responses, and querying this database to find matches for previously generated responses. How scalable is this approach in terms of storage space, compute, and accuracy? In this section we perform approximate calculations of these requirements on OpenAI's ChatGPT [Schulman et al., 2022].

**Storage space requirements**: We estimate ChatGPT's outputs to take 5TB space monthly (similar to a personal portable hard-disk) via the following calculations. ChatGPT currently gets about 2B monthly visits [NewYorkPost, 2023]. Assuming an average response length of 500 tokens per session, this corresponds to 1 trillion tokens. Similar in size to LLaMA's training data [Touvron et al., 2023], this needs 5TB space. However, 5TB is a small amount of storage compared to the industrial scale of information retrieval. For example, the Google Search index is over 100,000TB and has 100B+ pages [Google, 2023]. Major LLM service providers already have complex storage infrastructure to facilitate this defense. Additionally, OpenAI already stores conversations for at least 30 days (to monitor abuse, and potentially RLHF), even after a user chooses to opt-out [OpenAI, 2023b].

**Compute requirements**: Our retrieval experiments, conducted on a 14-core CPU (similar to a Macbook Pro), took 1 second per retrieval on a 15M sized corpus. Extrapolating to a corpus of ChatGPT's monthly usage (2B visits) would need 130 seconds/retrieval on a Macbook. However, this is fully parallelizable, and can make use of GPUs (Google searches 100B+ entries in < 1 sec). Moreover, efficient similarity search has powerful libraries like FAISS available [Johnson et al., 2019]. For comparison, ChatGPT itself takes 10 seconds/response, possibly using a powerful 8-GPU A100 server.[16] Major LLM providers have massive compute clusters, and we believe the computational requirement of retrieval is much lower than hosting LLMs in the first place, which these providers are already adept at. Moreover, our proposed ideas in Appendix A.3 can further reduce compute costs.

**Accuracy on larger databases**: Our experiments were conducted on the RankGen training set [Krishna et al., 2022a], which is the largest publicly available database of AI-generated text that we are aware of (15M generations each in four domains). Besides this, in Section 5 we also conducted experiments on a the ShareGPT corpus with 47K ChatGPT-generated responses. We note that it is extremely expensive and time consuming to create a corpus of AI-generated text from scratch: at a cost of $0.001 per 500-word response, collecting a billion ChatGPT outputs would cost $1M and take a long time to collect due to rate limits. Hence, a billion-scale experiment is likely only possible with an LLM provider's private database.

One of the concerns with a larger database is that of *semantic collisions*: a database will saturate with entries having similar semantics, especially for popular topics, and subsequently harm detection at scale. However, we note that:

1. Like other detectors, retrieval works best on longer sequences (Figure 6b). Long generations exponentially increase the likelihood of semantic divergences between pairs of entries.

---

[16]https://twitter.com/tomgoldsteincs/status/1600196981955100694

2. Retrieval compares the candidate response against the top-1 entry in the database, not the top-k. For false-positive candidates on popular topics, the top-k entries *together* are more likely to cover input semantics (recall) rather than top-1 (precision).

3. The most effective retrievers use a combination of neural semantic encoders and token overlap scores [Thakur et al., 2021]. We also see this our experiments (Section 5), BM25 beats P-SP at detection. BM25 is not purely semantic driven: it uses TF-IDF token overlap.

4. The retrieval accuracy for unperturbed AI-generated text is always 100%, just like exact match searches in a modern search engine. Retrieval-based detection is also effective on substrings of unperturbed text (as shown in Appendix G.2).

Overall, we are optimistic about our scaling plots (Figure 6a), and see just a 0.8% drop moving from a 1M to 10M database (PG19-BM25). We emphasize that BM25 is a basic retriever, and is not optimized on our task. Information retrieval literature has many powerful retrievers and has shown success at billion-scale corpus sizes [Gomes et al., 2013, Lakshman et al., 2021]. Google Search currently effectively indexes over 100B webpages [Google, 2023]. We have also suggested a dense retrieval mechanism in Appendix A.3 which can be optimized on the underlying retrieval corpus.

Retrieval can easily be used in tandem with other detectors like watermarking. Finally, we note that our paper is the first proof-of-concept that shows a retrieval-based detector could work, and we anticipate future work to build upon it.

## A.2 Limitations of retrieval for detection

While retrieval over previously-generated sequences is an effective defense against paraphrase attacks, it also suffers from key limitations, some of which apply broadly to all existing detectors. We discuss these limitations below and discuss possible solutions:

1. **Detection is specific to an API**. Unlike other general-purpose AI detection algorithms e.g. OpenAI's classifier [OpenAI, 2023a], retrieval can only detect generations from the API over which the database is built. API #1 has no access to the database of generations from API #2, and thus will not be able to detect generations produced by API #2.

2. **Retrieval is limited to closed-source LLMs**. Users of open-source LLMs like LLAMA [Touvron et al., 2023] can freely generate outputs without the outputs being stored in a central database. However, currently most major LLM providers operate their LLMs behind closed APIs. It is also important to note that watermarking [Kirchenbauer et al., 2023a], the most promising alternative to retrieval, also has this limitation. Since watermarks are added during decoding rather than into the model weights, users of open LLMs are free to generate text without watermarks. While other alternatives like DetectGPT [Mitchell et al., 2023] or classifiers do not suffer from this issue, we show that they either have low accuracy, or are extremely vulnerable to paraphrasing (Section 4.3).

3. **The API provider needs to provide a retrieval infrastructure**. After the release of ChatGPT [Schulman et al., 2022], AI chatbots are getting widespread adoption. At a conservative rate of 5M queries a day, the database will have almost two billion entries in a year. Complex retrieval infrastructure (like modern search engines) will be necessary to retrieve over these large databases with low latency.

4. **False positives due to training data memorization**. Language models have been shown to memorize sequences verbatim from their training data [Carlini et al., 2021], such as the Gettysburg Address [Radford et al., 2019]. Despite being originally written by humans, these sequences will be classified as model-generated by our detector. To tackle this issue, we suggest API providers additionally perform retrieval over the training data used to train the model. If a sequence is found in the training set as well as the generation database, it is likely to be an instance of training set memorization.

5. **Privacy concerns.** Providing a retrieval detection service partially exposes the database of previously generated text for *all* users. This raises concerns of membership inference attacks [Shokri et al., 2017] on private user data which may appear in the generated text (if present in user prompt). To mitigate this, we suggest: (1) the detection service should be provided only to trusted users under an agreement to not misuse the system, such as college teachers trying to detect cheating; (2) users should be encouraged not to enter any sensitive private data in their prompts to APIs, a

practice already followed by ChatGPT[17]; (3) API providers only provide a binary output from this detector (AI-generated or not), rather than actual search results; (4) API providers rate-limit queries from IP addresses; and (5) differential privacy mechanisms or scrubbing private attributes to make it difficult / impossible to reconstruct the user prompt from just detector access.

6. **Slight reduction in accuracy with large databases.** As we observed in Section 5.3, the accuracy of detecting paraphrased text slightly degrades as the database of retrievals gets larger. However, we found this decrease to be quite small (only 1% on PG19 scaling 1M generations to 15M), despite using fairly primitive retrievers like BM25. Moreover, unperturbed AI-generated text will always be detected with 100% accuracy using our method, irrespective of corpus size. We discuss this topic more in Appendix A.1 under "Accuracy on larger databases".

7. **Tasks with constrained output space or short outputs**. Similar to all other detection algorithms, it may be hard or even impossible to distinguish AI-generated outputs for tasks with a constrained output space (like sentence-level translation, classification) or very short outputs (as shown in Section 5.3). Thus, we believe the main utility of AI-generated text detection is for longer-form generated text, and hence we focus on tasks like long-form QA and open-ended text generation with relatively lengthy outputs. Note that to avoid detection, a sophisticated attacker may try to generate long-form text in smaller chunks using multiple API calls, where each newly-generated chunk is incrementally concatenated to the prompt. This is not a concern for our method if retrieval is done over the corpus of prompts concatenated with generations.

8. **Iterative attacks with access to detector.** Another concern is that attackers with access to detection algorithms will iteratively modify their perturbations until they avoid detection (as shown by Sadasivan et al. [2023]). While this is a valid concern for all detectors, we believe retrieval has an important advantage over the alternatives. Since the corpus of previously-generated text is proprietary, only the API provider can provide access to this detection service - it is impossible for attackers to locally reproduce this detector. This allows API providers to adopt several mitigation strategies such as (1) rate-limiting queries to avoid iterative attacks; (2) providing retrieval access only to verified users (e.g., teachers); and (3) detecting possible iterative attacks by analyzing previously queries to the retriever.

9. **Lack of formal guarantees between threshold $T$, response length and detection rate.** Unlike watermarking, we believe it is harder to establish formal relationships between the chosen threshold $T$, response length, and the corresponding TPR / FPR rates. Similar to DetectGPT / classifiers, we believe the threshold needs to be estimated empirically on the underlying corpus, retriever and candidate response distribution. A formal relationship between FPR and the threshold (like in watermarking) may be possible using information about the density of the retrieval database in the semantic vector space. We leave this exploration for future work.

### A.3   Ideas to make retrieval detection work well at an even larger scale

In Section 5.3, we observed that our proposed retrieval detector is effective even with a large corpus of 15M previously-generated sequences. While we do not have access to a larger corpus of generations (billion-scale), in this section we describe some ideas to improve retrieval detection at such a scale.

1. **Timestamp filtering in retrieval corpus.** To reduce the large search space, the detector interface could provide users with an option to restrict retrieval to only a fixed time period during which the text was likely to be generated. For instance, a common use-case of AI-generated text detection might be when teachers attempt to catch plagiarism in college essays. Teachers could restrict retrieval to only those generations created during the assignment window.

2. **More sophisticated retrieval strategies.** In our work, we only explore simple retrieval strategies like BM25. However, several more sophisticated retrieval strategies exist, which are known to boost performance [Thakur et al., 2021] and could be useful here. These include methods like re-ranking of top-$k$ retrievals [Khattab and Zaharia, 2020] or dense retrieval [Karpukhin et al., 2020]. We do note that these more complex methods are also slower, and latency is likely to be a pressing concern for API providers.

3. **Fine-tuning dense retrievers for the detection task.** The retrievers in our work are not fine-tuned for the task of AI-generated text detection. However, we hypothesize that fine-tuning

---

[17]https://chat.openai.com

retrievers on this task can help retrievers adapt better to the retrieval corpus and detection task. Specifically, a contrastive learning approach could be adopted here: positive pairs are paraphrased or otherwise noised sequences paired with their generations, while negative pairs are human-written continuations paired with the machine-generated text.

### A.4 Incentives for LLM providers to implement retrieval-based detection

Finally, in this section we discuss the incentive structures for LLM API providers, and why they might want to implement retrieval-based detection.

1. There is a substantial push from both the US and European governments to regulate companies to make their AI-generated text/images detectable. For instance, several major LLM providers made voluntary commitments to watermark their AI-generated content [Reuters, 2023a]. Similarly, the European government recently pushed for rulings to make AI-generated content detectable [Reuters, 2023b].

2. In a hypothetical scenario where there is a lawsuit about the origin of some malicious AI-generated content, maintaining a database of previously generated responses could be a reliable method to prove innocence, given its strong performance over competing AI-generated text detectors. Keeping this dataset private, also protects LLM providers from the privacy risks of retrieval-based detection (Appendix A.2).

3. Major LLM providers are already storing their model-generated outputs to monitor abuse, as well as possibly help improve their products with RLHF preference-based training. For instance, OpenAI's ChatGPT stores the history for 1 month even if users choose to opt-out [OpenAI, 2023b]. Hence, implementing a retrieval-based detection service on top of this database does not entail a resource overhead in terms of storage, and reduces the engineering effort needed to implement this detector.

**Is retrieval-based detection at odds with GDPR's right-to-be-forgotten?** One of the hurdles for LLM providers to implement retrieval-based detection is GDPR's right to be forgotten,[18] which allows attackers to request their data to be deleted by the LLM provider in order to avoid detection. However, we believe that AI-generated text detection is a sufficient cause to temporarily override GDPR deletion requests. We believe that AI-generated text detection could fall under the following GDPR guidelines for overrides: (1) "freedom of expression and information", (2) "establishment of a legal defense or in the exercise of other legal claims", and possibly (3) "comply with a legal ruling or obligation" in the future. As an example of this, while OpenAI allows users to delete their chat history [OpenAI, 2023b], they retain it for 30 days, and can review it if required to monitor for abuse.

## B  Ethical Considerations

Our goal in this paper is not to provide a recipe for potential attackers (e.g., college students wishing to use ChatGPT in their essays) to evade AI text detection systems. Rather, we wish to bring awareness to the wider community about the vulnerabilities of current AI-generated text detectors to simple paraphrase attacks. These detectors are not useful in their current state given how easy they are to evade. We encourage the research community to stress test their detectors against paraphrases, and to develop new detectors which are robust against these attacks. To facilitate such research, we open source our paraphraser and associated data / code.

Furthermore, we propose not just an attack but also a potentially strong defense against this attack. Our detection strategy is simple, relying on retrieval over a corpus of previously-generated sequences. We empirically show that such a detection algorithm could work at scale and provide extensive discussion on possible methods to improve performance (Appendix A.3), as well as discussing possible limitations and approaches to tackling them (Appendix A.2). We hope that retrieval-based AI-generated text detectors rapidly improve and are eventually deployed in conjunction with other detection methods like watermarking / classifiers.

---

[18] https://gdpr.eu/right-to-be-forgotten

# C   Experiments measuring intrinsic paraphrase generation quality

Our experiments in Section 4 and Section 5 focused on attacking AI-generated text detectors with paraphrases and defending against these paraphrase attacks. We used DIPPER as the underlying paraphrase generation model for all of these experiments. Are DIPPER's paraphrases actually good enough to make the attack worthwhile, and can simpler paraphrasers be just as effective as DIPPER? In this section, we conduct careful ablation experiments (Appendix C.1) and human evaluations (Appendix C.2) to validate the effectiveness of DIPPER at preserving the semantics of the input generation. Our results show that DIPPER effectively leverages surrounding context to paraphrase multiple sentences while preserving input semantics.

## C.1   Ablation studies on DIPPER

In this section, we perform automatic evaluations to confirm the efficacy of DIPPER as a paraphraser. From a survey of existing paraphrasers that we carry out in Appendix D.1, DIPPER possess two unique features that differentiate it from other paraphrasers: (1) its ability to leverage context from *outside* of the text to be paraphrased (such as the prompt); and (2) its ability to paraphrase multiple sentences at once. How useful are these features while paraphrasing long sequences of text?

To answer this question, we first train an ablated version of DIPPER by constructing a training dataset (Section 3) without any left or right context, and then fine-tuning T5-XXL using the same hyperparameters as in Section 3. We call this model DIPPER-no-ctx. We paraphrase 1K open-ended generations from GPT2-XL using both DIPPER and DIPPER-no-ctx, using each of the four configurations of diversity control codes studied in this paper. We then evaluate the quality of the paraphrased text using three metrics: (1) GPT3.5-davinci-003 perplexity [Brown et al., 2020] of the prompt concatenated with the paraphrased continuation; (2) RANKGEN compatibility between the prompt and the paraphrased continuation [Krishna et al., 2022a]; and (3) unigram token overlap between the paraphrased continuation and the prompt.

**Contextual paraphrasing leads to higher quality paraphrases**. In Table 3 (Experiment 1), we observe that across all four control code configurations and all three metrics, paraphrases from DIPPER are preferred over paraphrases from DIPPER-no-ctx. Specifically, with the lexical and order control codes set to 60% (most diverse), DIPPER paraphrases are preferred by GPT3.5 perplexity 71% of the time compared to non-contextual paraphrases (average perplexity drop of 12.9 vs 14.2).

**Paraphrasing multiple sentences at a time is better than paraphrasing individual sentences.** Next, we use our DIPPER-no-ctx model to compare two settings: paraphrasing 3 sentences at a time vs paraphrasing 1 sentence at a time before concatenating. We hypothesize that the former will produce higher quality paraphrases since we expect it to better connect discourse elements across the text. Indeed, in Table 3 (Experiment 2) across all control codes, GPT3.5 and RANKGEN usually prefer multi-sentence paraphrases over the single-sentence baseline. This preference is 71% or higher for all control codes when evaluating with GPT-3.5 perplexity, reaching 83% for L60,O60.

**DIPPER paraphrases are close to the unperturbed GPT-2 XL generations**. Finally, we compare DIPPER with the original GPT2-XL generations (without paraphrasing) on the same three metrics. While we expect metrics to prefer non-paraphrased text, a strong paraphraser will produce text that is close to the original in terms of these metrics. Table 3 (Experiment 3) confirms our hypothesis: at L20, RANKGEN has a 50-50 preference between the two outputs, while GPT3.5 prefers the non-paraphrased generations just 61% of the time, with an average perplexity gain of just 0.4 (11.1 to 11.5). At more diverse control codes, preference for GPT2-XL generations does go up (58% RANKGEN, 73% GPT3.5 for L60), but absolute scores continue to be close (11.1 vs 12.3 GPT-3.5 perplexity). Note that while all of these ablations use just a single paraphrase sample, it is easy for an attacker to obtain multiple samples from DIPPER and choose the sample that maximizes these metrics (as discussed in Section 4.3).

## C.2   Human evaluation of semantic preservation using DIPPER

The automatic semantic similarity scores in Table 1 and 3 indicate that DIPPER generates paraphrases that are faithful to the original input paragraphs. To confirm this result with human evaluation, we

Table 3: Ablation experiments demonstrate the high quality of DIPPER's paraphrases compared to alternatives. Displayed scores are the percentage of cases in which rewrite A is preferred over B by one of the three metrics, with subscripts showing absolute average scores on each metric across the dataset. Overall, DIPPER benefits from context outside the input (Experiment 1), multi-sentence paraphrasing (Experiment 2), and is not too far behind non-paraphrased text in terms of quality (Experiment 3).

| | **Open-ended generation with GPT2-XL on Wikipedia prompts** | | | | | |
| --- | --- | --- | --- | --- | --- | --- |
| | RANKGEN-XL | | GPT3.5 davinci-003 perplexity | | unigram overlap with prompt | |
| Control | rewrite A | rewrite B | rewrite A | rewrite B | rewrite A | rewrite B |
| **Experiment 1**: *Is context helpful for paraphrasing?* | | | | | | |
| rewrite A = DIPPER with context | | | | | | |
| rewrite B = DIPPER no context | | | | | | |
| 20L | **62**% $_{10.2}$ | 38% $_{9.4}$ | **58**% $_{11.5}$ | 42% $_{11.7}$ | **55**% $_{41.3}$ | 45% $_{40.7}$ |
| 40L | **62**% $_{9.8}$ | 38% $_{8.7}$ | **64**% $_{11.9}$ | 36% $_{12.5}$ | **57**% $_{40.7}$ | 43% $_{39.8}$ |
| 60L | **64**% $_{9.6}$ | 36% $_{7.9}$ | **66**% $_{12.3}$ | 34% $_{13.3}$ | **55**% $_{39.9}$ | 45% $_{39.2}$ |
| 60L,60O | **66**% $_{8.3}$ | 34% $_{6.3}$ | **71**% $_{12.9}$ | 29% $_{14.2}$ | **56**% $_{39.4}$ | 44% $_{38.4}$ |
| **Experiment 2**: *Is it helpful to paraphrase multiple sentences at a time?* | | | | | | |
| rewrite A = DIPPER 3 sentences at a time | | | | | | |
| rewrite B = DIPPER 1 sentence at a time | | | | | | |
| 20L | **55**% $_{9.4}$ | 45% $_{8.9}$ | **73**% $_{11.7}$ | 27% $_{12.5}$ | **51**% $_{40.7}$ | 49% $_{40.7}$ |
| 40L | **55**% $_{8.7}$ | 45% $_{8.5}$ | **71**% $_{12.5}$ | 29% $_{13.3}$ | 46% $_{39.8}$ | **54**% $_{40.3}$ |
| 60L | **53**% $_{7.9}$ | 47% $_{7.6}$ | **71**% $_{13.3}$ | 29% $_{14.3}$ | 46% $_{39.2}$ | **54**% $_{39.8}$ |
| 60L,60O | **57**% $_{6.3}$ | 43% $_{5.3}$ | **83**% $_{14.2}$ | 17% $_{16.7}$ | 47% $_{38.4}$ | **53**% $_{38.9}$ |
| **Experiment 3**: *Does paraphrasing preserve the quality of the original text?* | | | | | | |
| rewrite A = no paraphrasing | | | | | | |
| rewrite B = DIPPER | | | | | | |
| 20L | **50**% $_{10.4}$ | **50**% $_{10.2}$ | **61**% $_{11.1}$ | 39% $_{11.5}$ | **51**% $_{41.6}$ | 49% $_{41.3}$ |
| 40L | **57**% $_{10.4}$ | 43% $_{9.8}$ | **67**% $_{11.1}$ | 33% $_{11.9}$ | **55**% $_{41.6}$ | 45% $_{40.7}$ |
| 60L | **58**% $_{10.4}$ | 42% $_{9.6}$ | **73**% $_{11.1}$ | 27% $_{12.3}$ | **58**% $_{41.6}$ | 42% $_{39.9}$ |
| 60L,60O | **68**% $_{10.4}$ | 32% $_{8.3}$ | **79**% $_{11.1}$ | 21% $_{12.9}$ | **61**% $_{41.6}$ | 39% $_{39.4}$ |

hire three native English teachers and/or editors on Upwork[19] to evaluate the semantic fidelity of the paraphrases. As human evaluation is expensive, we fix the order diversity ($O$) to be 0 and focus on the impact of the lexical diversity. We evaluate paraphrases with the lexical codes $L20$, $L40$, and $L60$, corresponding to moderate, medium, and high lexical diversity. Twenty paraphrases are sampled randomly for each lexical code, resulting in 60 original text and paraphrase pairs.

The evaluation is conducted on the platform Label Studio [Tkachenko et al., 2020-2022].[20] As shown in the interface of our annotation platform Figure 7, the text to be paraphrased (highlighted in yellow) are preceded by its context. The annotators see the same amount of text as DIPPER. They need to first read the texts, select one point on the Likert scale, then provide free-form comments justifying their ratings. We estimated that the evaluation of each paraphrase takes 1.5 to 2 minutes. As such, we pay $15 as a base rate with a bonus for the reasonable extra time that the annotators spend on the tasks.

Among the 60 original text and paraphrase pairs, the three annotators agreed on their choice 28.3% of the time, and 60% of the time the point they chose on the scale differs by 1. Table 4 reports how often each point on the Likert scale is chosen. Over 80% of the time, our annotators rate DIPPER's paraphrases as nearly equivalent (4 out of 5) or approximately equivalent (5 out of 5).

A qualitative analysis of the free-form annotator comments reveals systemic strengths and shortcomings of DIPPER. Table 8 provides two representative examples for each lexical code that is evaluated in our human study.

---

[19] https://www.upwork.com
[20] https://labelstud.io/

Figure 7: The interface of the annotation platform used in our human study

Table 4: This table shows how often each point in the Likert scale was chosen by 3 annotators for the pairs of original and paraphrased texts. Twenty text pairs are randomly selected for each lexical code (L). 81.8% of the time, our model DIPPER provides a paraphrase which is nearly equivalent to the input in terms of semantic meaning.

| L | Sum of 4 and 5 | 5 Approx. equivalent | 4 Nearly equivalent | 3 Somewhat equivalent | 2 Topically related |
|---|---|---|---|---|---|
| 20 | 95.0% | 63.3% | 31.7% | 5.0% | 0.0% |
| 40 | 78.3% | 45.0% | 33.3% | 21.7% | 0.0% |
| 60 | 70.0% | 28.3% | 41.7% | 28.3% | 1.7% |
| **Total** | 81.1% | 45.6% | 35.6% | 18.3% | 0.6% |

**Strengths** First, the third example in Table 8 exemplifies DIPPER's ability to leverage information from context to increase diversity while maintaining coherence (i.e., from *line. . . reference the song's title* to *reference to "I'm the Greatest"*). The same is observed in row 2 where DIPPER uses the context to interchange *he* and *Churchill*. A paraphrase model without looking into context will have great difficulty in doing this and no prior paraphraser (see Table 5 for a list) is capable of that. Second, the example in the fifth row highlights DIPPER's ability to make significant changes to original texts with a high lexical diversity code ($L60$) (see the color coding) while preserving their semantic meaning as rated by the annotators.

**Qualitative shortcomings**: The first shortcoming is that, when the original text contains new created proper names (unlike common people and country names), such as the ones in row 6 (*Homing Attack* and *Slide Attack*), a high lexical code has a tendency to change such nouns, leading to the result that one of our annotators deems it to be only topically related to the original. However, this shortcoming can be overcome by decreasing the lexical code, which a user can choose from a continuous range (from 0 to 100). For instance, in row 1 with `lex=20`, the songs' names *M's Confession* and *Gone Fishing* are kept intact. Another shortcoming is that DIPPER occasionally omits content from an original text. While in some cases such removal is acceptable (see row 6), in other cases it causes significant change in the meaning of the text (see row 4). However, the former case can be overcome by paraphrasing a shorter paragraph at a time.

Overall, the human study shows that DIPPER performs well at preserving the semantic meaning of original texts while introducing both semantic and syntactic diversity. Because DIPPER provides user-friendly controllabilty of output diversity, a user can adjust the control code to find the most suitable paraphrase for their need.

## D   Related work for discourse paraphrasing

### D.1   Survey of paraphrase generation papers

As an important NLP task, paraphrasing has attracted much attention. Many models have been proposed to improve the quality of paraphrases. To position our model DIPPER and highlight its strengths, we conduct a survey of paraphrase generation papers from 2018 to 2022 (Table 5) and focus on the following four aspects:

1. Whether a model can paraphrase a paragraph at once,
2. whether a model can merge or split consecutive sentences when appropriate,
3. whether a model leverages context surrounding an input sentence when paraphrasing,
4. whether a model provides control knobs for users to customize the output diversity.

The survey shows that only three out of 25 papers mentioned that their model can paraphrase more than one sentence (but not necessarily at once). None of them enables their model to merge or split sentences when paraphrasing. No model uses information from context surrounding an input sentence during inference time. Finally, 14 papers offer ways for users to customize the diversity of paraphrases. However, most diversity control methods such as constituency parses or exemplars may not be straightforward and intuitive to end-users as the scalar control knobs in DIPPER.

In contrast to the papers in the survey, DIPPER nicely combines all desiderata into one model and offers intuitive control knobs for lexical and syntactic diversity. Automatic and human evaluation show that DIPPER can efficiently leverage context information and reorganize sentences while having high fidelity in meaning (Appendix C).

### D.2   Other related work

In this section we discuss a few additional less related papers which were not included in our survey in Appendix D.1. Our discourse paraphraser is closely related to work on contextual machine translation, where source/target context is used to improve sentence-level machine translation [House, 2006, Jean et al., 2017, Wang et al., 2017, Tiedemann and Scherrer, 2017, Kuang et al., 2018, Agrawal et al., 2018, Miculicich et al., 2018, Zhang et al., 2018, Xiong et al., 2019, Jean et al., 2019, Voita et al., 2019a, Yin et al., 2021, Mansimov et al., 2021]. Prior work has shown that context helps with anaphora resolution [Voita et al., 2018], deixis, ellipsis, and lexical cohesion [Voita et al., 2019b]. Efforts to make paraphrase generation more contextual have been quite limited. A few efforts have attempted to use sentence level context to paraphrase phrases [Connor and Roth, 2007, Max, 2009], and dialogue context to paraphrase individual dialogues in a chat [Garg et al., 2021].

Our work is also related to efforts in text simplification to go beyond a sentence, by collecting relevant datasets [Xu et al., 2015, Devaraj et al., 2021] and building unsupervised algorithms [Laban et al., 2021]. Note that our work focuses on a general-purpose paraphrasing algorithm and is not tied to any particular style, but could be utilized for document-level style transfer using techniques like Krishna et al. [2020, 2022b]. Similar efforts have also been undertaken in machine translation, [Popescu-Belis et al., 2019, Junczys-Dowmunt, 2019, Maruf et al., 2021], attempting to translate paragraphs/documents at once.

## E   More background on detectors of AI-generated text

In this section, we provide an overview of existing algorithms that have been developed for the purpose of detecting machine-generated text. Such algorithms fall into three main categories: (1) watermarking algorithms, which modify the generative algorithm to encode hidden information unique to the API (Appendix E.1); (2) statistical outlier detection methods, which do not modify

Table 5: The table shows the result of our survey of paraphrase generation papers from 2018 to 2022. We focus on four aspects: (1) whether a model can paraphrase multiple sentences at once, (2) whether a model is able to merge or split an input sentence when appropriate, (3) whether a model takes context surrounding the input sentence into consideration when paraphrasing, and (4) whether a model enables users to control the semantic and syntactic diversity of paraphrases. [1]Granularity levels are *word*, *phrase*, and *sentence*. [2]Meng et al. [2021] use context for their dataset construction, but do not leverage it during training/inference. [3]The diversity score is a combination of the unigram Jaccard distance and the relative position change for unigrams. [4]The code is represented by a three dimensional vector corresponding to semantic similarity as well as syntactic and lexical distances between the input and output sentences.

| Paper | Multi-sentence | Merge / Splits | Contextual | Diversity Control |
|---|---|---|---|---|
| Iyyer et al. [2018] | ✗ | ✗ | ✗ | Constituency parse |
| Li et al. [2018] | ✗ | ✗ | ✗ | ✗ |
| Roy and Grangier [2019] | ✗ | ✗ | ✗ | ✗ |
| Witteveen and Andrews [2019] | ✓ | ? | ✗ | ✗ |
| Kumar et al. [2019] | ✗ | ✗ | ✗ | ✗ |
| Hu et al. [2019] | ✗ | ✗ | ✗ | Decoding constraints |
| Chen et al. [2019] | ✗ | ✗ | ✗ | Exemplar |
| Li et al. [2019] | ✗ | ✗ | ✗ | Granularity control[1] |
| Goyal and Durrett [2020] | ✗ | ✗ | ✗ | Exemplar |
| Lewis et al. [2020] | ✓ | ? | ✗ | ✗ |
| Thompson and Post [2020] | ✗ | ✗ | ✗ | $n$-gram overlap |
| Kumar et al. [2020] | ✗ | ✗ | ✗ | Exemplar |
| Kazemnejad et al. [2020] | ✗ | ? | ✗ | ✗ |
| Krishna et al. [2020] | ✗ | ✗ | ✗ | ✗ |
| Rajauria [2020] | ✗ | ✗ | ✗ | ✗ |
| Meng et al. [2021] | ✗ | ✗ | ✗[2] | Diversity score[3] |
| Huang and Chang [2021] | ✗ | ✗ | ✗ | Constituency parse |
| Lin et al. [2021] | ✓ | ✗ | ✗ | ✗ |
| Goutham [2021] | ✗ | ✗ | ✗ | ✗ |
| Damodaran [2021] | ✗ | ✗ | ✗ | Binary |
| Dopierre et al. [2021] | ✗ | ✗ | ✗ | $n$-gram |
| Bandel et al. [2022] | ✗ | ✗ | ✗ | Control code[4] |
| Hosking et al. [2022] | ✗ | ✗ | ✗ | Syntactic sketch |
| Yang et al. [2022] | ✗ | ✗ | ✗ | Examplar+Keywords |
| Xie et al. [2022] | ✗ | ✗ | ✗ | ✗ |
| **DIPPER (ours)** | ✓ | ✓ | ✓ | ✓ |

the generative algorithm but look for inherent artifacts in generated text (Appendix E.2); and (3) classifiers trained to discriminate machine-generated text from human-written text (Appendix E.3). Finally, in Appendix E.4, we compare and contrast our work to Sadasivan et al. [2023], who also note the efficacy of paraphrasing attacks but do not consider a retrieval-based defense in their pessimistic conclusion about the fate of AI-generated text detection.

### E.1 Watermarking language model outputs

A "watermark" is a modification to the generated text that can be detected by a statistical algorithm while remaining imperceptible to human readers. Effective watermarks are difficult to remove and have little effect on the quality of generated text. Prior work attempted to watermark natural language using syntax tree manipulations [Topkara et al., 2005, Meral et al., 2009], and this area has gotten renewed interest with large language models generating human-like text [Abdelnabi and Fritz, 2021, Grinbaum and Adomaitis, 2022]. Most recently, Kirchenbauer et al. [2023a] propose a simple algorithm that only requires access to the LLM's logits at each time step to add watermarks. The watermark can then be verified with only blackbox access to the LM and knowledge of a specific hash function. This algorithm operates in three steps:

1. **Mark a random subset of the vocabulary** as "green tokens" (or tokens representing the watermark, as shown in Figure 1) using the hash of the previously generated token as a random seed. A total of $\gamma|V|$ tokens are marked green where $\gamma$ is the fraction of the tokens that are watermarked with default $\gamma = 0.5$.

2. **Increase the logit value** for every green token by a constant $\delta$ ($= 2$ by default), which denotes the watermark strength. This raises the probability of sampling green watermarked tokens, especially for high-entropy distributions.

3. **Sample sequences** using decoding algorithms such as nucleus sampling [Holtzman et al., 2020], leveraging the modified probability distribution at each timestep before truncation.

**Detecting the watermark**: Verifying whether a text is generated by a watermarked LM is possible with just knowledge of the hash function and tokenizer. Specifically, the verifier tokenizes the text and counts the number of green tokens it contains. This is used to calculate the standard normal score ($z$-score) for the hypothesis test. If the sequence with $T$ tokens contains a certain number of the green token (denoted as $|s|_G$), the $z$-score can be computed by:

$$z = (|s|_G - \gamma T)/\sqrt{T\gamma(1-\gamma)}$$

Intuitively, a higher $z$-score implies it is less likely for a human to have written the text (null hypothesis) since it contains a higher than expected number of green tokens. Kirchenbauer et al. [2023a] recommend using a high $z$ value ($z > 4$, or $p < 3 \times 10^{-5}$) to reduce the risk of false positives (human-written text classified as AI-generated). Low false positive rates are critical in AI-generated text detection algorithms [OpenAI, 2023a]—we discuss this in Section 4.1.

## E.2 Statistical outlier detection methods

Unlike the watermarking algorithms, outlier detection algorithms make no modification to the generative algorithm. Instead, they attempt to distinguish between human-written and machine-generated text based on the presence of artifacts in generated text [See et al., 2019, Holtzman et al., 2020]. Early methods detect statistical irregularities in measures such as entropy [Lavergne et al., 2008], perplexity [Beresneva, 2016], and $n$-gram frequencies [Grechnikov et al., 2009, Badaskar et al., 2008]. After the release of GPT-2, Gehrmann et al. [2019] introduced the GLTR visualization tool to assist human verifiers in detecting machine-generated text. Most recently, the release of ChatGPT has prompted the development of two new tools, namely a closed-source tool called GPTZero [Tian, 2023], and open-source DetectGPT [Mitchell et al., 2023]. DetectGPT uses an observation that model-generated text lies in the negative curvature regions of the model's log probability function. It constructs multiple perturbations of the model generated text (using a mask-and-fill strategy), and compares the log probability of the perturbations with the unperturbed generation. Text is considered model generated if the log probability of the unperturbed text is significantly higher than the log probability of perturbations.

## E.3 Classifiers

The third class of detection methods relies on classifiers that are fine-tuned to distinguish human-written text from machine-generated text. Early efforts in this vein use classifiers to detect fake reviews [Hovy, 2016] and fake news [Zellers et al., 2019]. Other related studies examine classification performance across domains [Bakhtin et al., 2019] and decoding strategies [Ippolito et al., 2020]. Such studies inspired others to use their insights to improve generative performance [Deng et al., 2020, Krishna et al., 2022a]. Most recently, OpenAI fine-tuned a GPT model to perform this discrimination task and released it as a web interface [OpenAI, 2023a]. They fine-tuned this classifier using generations from 34 language models, with text sourced from Wikipedia, WebText [Radford et al., 2019], and their internal human demonstration data.

## E.4 Comparison to Sadasivan et al. (2023)

In very recent concurrent work, Sadasivan et al. [2023] also demonstrate the utility of paraphrasing attacks against AI-generated text detectors. While their work makes use of off-the-shelf sentence-level paraphrase models, DIPPER possesses advanced discourse-level rewriting capabilities as well

as fine-grained diversity control, which allows us to thoroughly analyze the effectiveness of various paraphrasing strategies. Our experiments also encompass more tasks, datasets, and detection algorithms. Moreover, we evaluate larger language models like GPT3.5-davinci-003. Finally and most importantly, our retrieval-based defense *directly contradicts* the "impossibility result" of Sadasivan et al. [2023] and its associated proof, which states that even an optimal detector will approach the performance of a random classifier as the distance between the distributions of LLM-generated text and human generated text goes to zero. Since our detector does not rely on properties of the text but rather a corpus search, the quality of the generated text is irrelevant to the effectiveness of our detector, and thus their proof does not apply to our method.

# F   More experimental details of our attack experiments

## F.1   Details for training our paraphraser DIPPER

Our paraphraser DIPPER is a sequence-to-sequence Transformer neural network [Vaswani et al., 2017], initialized with the T5-XXL 1.1 checkpoint [Raffel et al., 2020] and fine-tuned on our paraphrase generation data, using early stopping on validation loss for held-out novels. During training, we find it helpful to paraphrase a maximum of 3 consecutive sentences at time, which leads to better adherence to control codes. Our models are implemented in JAX [Bradbury et al., 2018] using the T5X library [Roberts et al., 2022] with the default fine-tuning hyperparameters. Our final dataset contains 6.3M paraphrase pairs. Training was done on 64 cloud TPUv3 chips, and took 6-12 hours to complete. At inference time, we use nucleus sampling [Holtzman et al., 2020] with $p = 0.75$ and a variety of control codes.

To make our paper more intuitive, we have slightly modified the notation that our actual pretrained model uses. Our pretrained model uses control codes $100 - L$ and $100 - O$, denoting lexical/order *similarity* rather than diversity. Also, `<sent>` is used instead of `<p>`. We will clearly document this in the code release.

## F.2   Long-form question answering data processing

In Section 4 evaluate long-form question answering [Fan et al., 2019], in which an LM must answer a how/why question (e.g., *Why are almost all boats painted white?*) with a 250-350 word answer. To build a long-form question answering dataset, we scrape questions from the r/explainlikeimfive subreddit posted between July to December 2021.[21] We randomly sample 500 questions from each of six popular domains on the subreddit (biology, physics, chemistry, economics, law, and technology) and pair each question with its longest human-written answer, which yields 3K long-form QA pairs.

## F.3   Are successful attacks against model-specific detectors really an attack success?

In our experiments, we attack two model-specific detectors (watermarking, DetectGPT) in addition to model-agnostic detectors (OpenAI classifier, GPTZero, RankGen). Model-specific detectors have been specifically designed to judge whether a response was generated by a *particular* model. After paraphrasing, a response is not truly generated by that specific model anymore—it has been generated by the *paraphrasing* model. Hence, in a sense, the inability of these detectors to detect paraphrased text denotes their robustness in model-specific detection.

While the argument above has merit, we argue that it is critical for model-specific detectors to be robust against paraphrasing attacks. Given their strong performance over model-agnostic methods (Section 4.3), and the large risk of perturbation attacks (human-edited or automatically-edited), we think it is important for model-specific detectors to bake perturbation robustness in their design for better downstream usability. Currently, the low performance of model-agnostic detectors (Section 4.3) makes them quite unusable, and OpenAI even took down their text classifier due to low performance [OpenAI, 2023a]. Currently, model-specific detectors (including watermarking and retrieval) seem to be the only reliable path towards robust AI-generated text detection.

---

[21]We choose this period since current language models have been trained on internet data available before June 2021 [OpenAI, 2022], this prevents verbatim copying from training data.

# G   More retrieval-based detection experiments

## G.1   Controlled comparisons of retrieval with other AI-generated text detectors on open-ended text generation

We conduct a controlled comparisons of retrieval on the open-ended text generation task with Wikipedia prompts (see Section 5.2). The result of the experiment is presented in Table 7.

## G.2   Robustness against text shortening and mixing attacks

Our experiments in Section 5 assumed that an attacker would paraphrase the entire LLM output to evade detection. However, an alternative attack strategy that might be adopted in practice is *text shortening*, or *text mixing* [Kirchenbauer et al., 2023b]. Here, attackers only use a substring of an LLM generated output (instead of the full output), and optionally mix it with text from other sources (like other LLM-generated outputs or human-written text).

In this section, we conduct some preliminary experiments to show the robustness of retrieval-based detection to text shortening and mixing attacks. We adopt the same experimental setup as Section 5.3, utilizing a BM25 retrieval on a PG19 machine-generated corpus of 2M responses.

In Table 6 we see that retrieval is a promising defense strategy even against text shortening and mixing attacks. Overall, we see that unperturbed random substrings (from single or multiple generations) can still be detected quite easily (86.2% to 94.7% detection rate at 1% FPR). However, adding DIPPER paraphrasing on top of that reduces accuracy (56.1% to 68.8% detection rate).

Table 6: Preliminary experiments measuring the robustness of retrieval-based detection to LLM response shortening and mixing attacks. Overall, we find that retrieval is a strong detector for truncated as well as mixed responses. Experiments are conducted with BM25 retrieval in the PG19 data setup with a retrieval corpus of size 2M responses (from Section 5.3). Results shown are true positive rates at the 1% FPR threshold.

| Candidate Text | Retrieval detection rate (1% FPR) |
| --- | --- |
| unperturbed LLM response | 100.0 |
| DIPPER-paraphrased response | 98.2 |
| 50% truncated DIPPER-paraphrased response | 72.6 |
| 50% truncated LLM response | 86.2 |
| 50% LLM response 1 + 50% LLM response 2 | 94.7 |
| 50% DIPPER-paraphrased response | 68.8 |
| 50% DIPPER-paraphrased response 1 + DIPPER-paraphrased response 2 | 56.1 |

# H   ROC curves at different FPR

See Figure 8.

Table 7: Our retrieval defense significantly improves AI-generated text detection accuracy (at 1% FPR) over baselines on all settings, including our most diverse paraphrase attacks (+60L and +60L,60O).

| | **Open-ended text generation with Wikipedia prompts** (300 generated tokens) | | | | | | | | |
| | GPT2-XL | | | OPT-13B | | | GPT-3.5 (davinci-003) | | |
| | Original | + 60L | + 60L,60O | Original | + 60L | + 60L,60O | Original | + 60L | + 60L,60O |
| *Baseline methods*: | | | | | | | | | |
| Watermark | 100.0 | 68.9 | 57.2 | 99.9 | 63.7 | 52.8 | - | - | - |
| DetectGPT | 70.3 | 8.7 | 4.6 | 14.3 | 0.8 | 0.3 | 2.0 | 0.5 | 0.0 |
| OpenAI | 21.6 | 13.3 | 14.8 | 11.3 | 9.1 | 10.0 | 30.0 | 15.6 | 15.6 |
| *(Ours)* Retrieval over corpus of 3K generations from model itself, with retriever: | | | | | | | | | |
| SP | 100.0 | 86.4 | 81.5 | 100.0 | 84.4 | 77.7 | 100.0 | 65.9 | 49.5 |
| BM25 | 100.0 | 99.0 | 98.0 | 100.0 | 97.2 | 95.3 | 100.0 | 58.8 | 37.4 |
| *(Ours)* Retrieval over corpus of 9K generations pooled from all three models, with retriever: | | | | | | | | | |
| SP | 100.0 | 72.1 | 63.2 | 100.0 | 74.6 | 65.6 | 100.0 | 63.1 | 45.6 |
| BM25 | 100.0 | 85.0 | 78.7 | 100.0 | 87.2 | 79.1 | 100.0 | 58.8 | 37.4 |

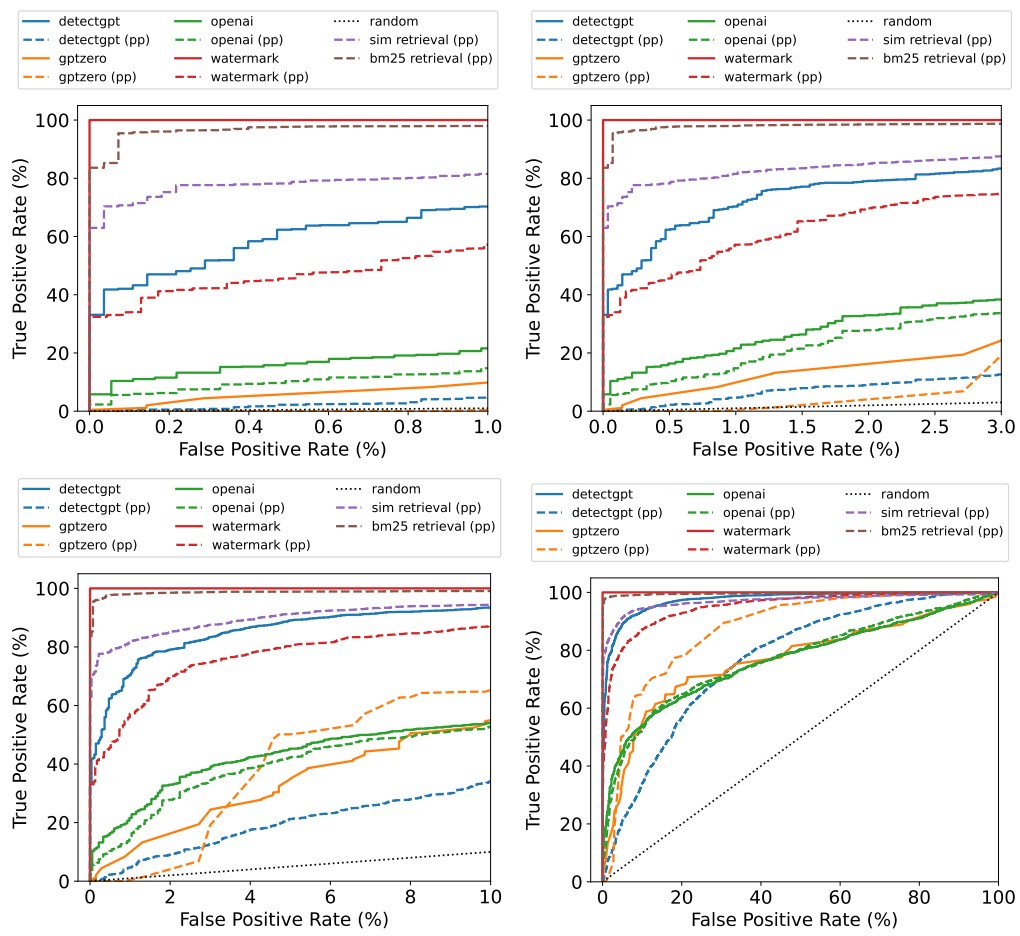

Figure 8: ROC curves for text generated by GPT2-XL, before paraphrasing (solid lines) and after paraphrasing (dashed lines, pp). Different plots represent different clipping thresholds on the X-axis.

Table 8: Representative model outputs of each lexical code with Likert ratings and comments from the annotators. The texts in bold in the original texts are the context. Red words are the content being changed in the original text and green words are the changed content in the paraphrases.

| L | Original | Paraphrase | Annotator Comment |
|---|---|---|---|
| 20 | **The first, originally used as Renard's theme, is heard during the mammoth "Antonov" cue on the recording, and is written for piano. The second new theme, used in the "Christmas in Turkey" track of The World Is Not Enough, is reused in the "Going Down Together" track.** Both themes were later adapted by David Arnold and featured in the film's score. The first theme appears in the cue "M's Confession" and the second theme appears in the cue "Gone Fishing". | Both themes were subsequently adapted by David Arnold and used in the film score. The first theme is used in the "M's Confession" cue, and the second in the "Gone Fishing" cue. | **5—Approx. equivalent** While the paraphrase is worded in a more simplistic way than the original and has shortened clauses, the information presented in both is the same. |
| 20 | **Churchill argues that the United States instituted blood quantum laws based upon rules of descendancy [. . . ]** He has argued that the United States has used them to strip Indigenous people of their rights, land and culture in order to benefit those in power. He contends that these laws were designed to create a divide among Indigenous people, making them easier to control and manipulate. Churchill has further argued that this manipulation continues today, as Indigenous people are often forced to deny their heritage in order to have access to resources and opportunities that are denied to them as a result of their heritage. | He has argued that the United States has used them to strip indigenous people of their rights, land, and culture in order to benefit those in power. Churchill also argues that these laws were designed to create divisions among indigenous people in order to make them easier to control and manipulate. He has also argued that this manipulation continues today, in that indigenous people are often forced to deny their heritage in order to have access to resources and opportunities that are denied to them because of their heritage. | **5—Approx. equivalent** The paraphrase retains all the information of the original, but uses synonymous language in several places. Also, the paraphrase replaces the pronoun "he" with "Churchill" and vice versa. |
| 40 | **When tailoring his 1970 composition "I'm the Greatest" for Starr to record on the 1973 album Ringo, Lennon referenced the song title with the lines [. . . ]** Lennon's original line was changed to "Back off Boogaloo" to reference the song's title, creating the line that Starr sang in the finished version. | Lennon's original reference to "I'm the Greatest" was changed to "Back off Boogaloo" to create the line Starr sang on the finished recording. | **4—Nearly equivalent** The original does not specify that the line in question was "I'm the Greatest." However, this can be inferred from the rest of the passage. Otherwise, the paraphrase and the original are quite similar. |
| 40 | **Some monuments were vandalized and some were moved to new locations. [. . . ]** In some cases, the monuments were moved to new locations. For example, the stairway of the Great Pyramid of Cholula in Mexico was relocated in the 19th century and then returned to its original location in the 20th century. Other monuments were destroyed during wars or religious conflicts, such as the Buddhas of Bamiyan in Afghanistan, which were destroyed by the Taliban in 2001. | In some cases, monuments were moved to new locations. The Great Pyramid of Cholula, in Mexico, was moved in the 19th century and then returned to its original place in the 20th. Others were destroyed during wars or religious controversies, such as the Buddhas of Bamiyan in Afghanistan, which were destroyed by the Taliban in 2001. | **3—Somewhat equivalent** Only one major change in this short text, but I feel it's significant. The original text speaks of the moving of a pyramid's staircase, but the paraphrased text implies that the entire pyramid was moved and then replaced. This massively changes the meaning of this text. |
| 60 | **Heaving wildly, the Hunyani began to descend rapidly. [. . . ]** The crew was unable to regain control and the Hunyani crashed into the ground at 17:14, killing all onboard. The crash was investigated by the Civil Aviation Authority of Zimbabwe and the investigation concluded that the crash was caused by a combination of pilot error, fatigue, and weather conditions. The pilot had failed to properly assess the weather conditions before takeoff, instead relying on a forecast from a private weather service. The weather conditions worsened as the plane flew and with the pilots already fatigued, they were unable to appropriately react to the changing conditions. The combination of these factors led to the plane losing control and crashing. | The crew could not regain control, and at 17:14 the Hunyani plunged into the ground and all those on board were killed. The accident was investigated by the Civil Aviation Authority of Zimbabwe, which found that the crash was caused by a combination of pilot error, fatigue and the weather. The pilots had not properly examined the weather before take-off, and instead had relied on a forecast from a private meteorological service. The weather deteriorated during the flight, and the exhausted pilots were unable to react properly to the deteriorating conditions. | **5—Approx. equivalent** Well chosen words make the alternate version different but retain the original meaning well. The only difference is that the paraphrased version does not have the concluding sentence, but it does not alter the meaning in any way as the facts have already been included previously. |
| 60 | **Sonic's main means of attack is the Spin Attack, in which he curls into a ball and rapidly spins his body, damaging enemies and certain obstacles upon collision. This may be performed by jumping or by rolling on the ground.** Sonic can also use a variety of other moves, such as the Homing Attack, Slide Attack, and Wall Jump. He also has access to power-ups like the Speed Shoes, Invincibility, and the Super Peel Out. In some games, Sonic can also use special items or vehicles to traverse levels. | Also, the character can use a variety of other attacks, such as the homing attack, the slide attack, and the wall jump. In addition, he can acquire power-ups such as the Speed Boost, Super Speed, and the invincibility meter, and in certain games, he may make use of special items or vehicles to traverse the world. | **2—Topically related** In the second part of the paraphrase, the writer ignores the actual moves and skills of Sonic and invents some entirely different ones that not mentioned in the original text. The method of performing the attack moves has also been missed out. |

