# OpenReview forum: "Paraphrasing evades detectors of AI-generated text, but retrieval is an effective defense"
_NeurIPS.cc/2023/Conference — NeurIPS 2023 poster_

### Official Review · Reviewer_44uy · 2023-06-30

**Soundness:** 2 fair
**Presentation:** 4 excellent
**Contribution:** 3 good
**Rating:** 6
**Confidence:** 4

**Summary:**

The paper can be split into two parts. In the first part, the authors propose a paraphrasing-based attack that circumvents various AI-generated text detectors. The authors introduce DIPPER, an 11B parameters Transformer model obtained by fine-tuning T5-XXL. By using DIPPER to create paraphrases from texts generated by LLMs, the authors show that they can successfully evade various commonly-used detectors. In the second part, the authors propose a retrieval-based generated text detector that can successfully detect instances of paraphrased generated text.


**Strengths:**

- The paper is well-written and easy to follow.

- The authors introduce DIPPER, a paraphrasing model based on T5-XXL. The authors show that DIPPER can generate high-quality paraphrases and have tested the model in a human trial included in the Supplementary materials.

- The paraphrasing attacks are effective against the models tested.

- The authors have an extensive discussion about the limitations of their retrieval-based system.


**Weaknesses:**

- The paper's biggest weakness is the generated text detection via retrieval methods since many assumptions must be made for it to work correctly.

  - First, one needs access to all of the text generated by the model we want to test against. This is significantly limiting since the data would probably be available only to the entity that is maintaining the LLM API. Maintaining such a big database of generated text would also be expensive from a storage perspective, and computationally expensive for the retrieval phase.

  - Second, using a similarity score to compare the paraphrased text with retrieved texts, one must assume that the similarity between the generated text and its paraphrased variant would be high. However, the similarity score could also be large if the two texts have similar semantics. This would be true if one used something similar to the cosine similarity between two text embeddings obtained by a neural encoder [[1]]. If one maintains a significantly large and diverse database of generated text, the chance of it containing an entry that is semantically similar to a candidate text grows, leading to FP predictions.

  - Third, the method would only be viable for closed, proprietary models, and would not apply to open-source models, since it is impossible to collect a database of all the text generated by an open-source LLM that is hosted by multiple people and institutions.

- As far as I understand, the authors have used human-generated continuations only to adjust the detection thresholds to maintain a 1% FPR (lines 171-172). If this is the case, the evaluation methodology is somewhat flawed due to the points I've previously made. In my opinion, a better evaluation methodology would be using some parts of the human-generated data to fix a low FPR, and also have human-generated data with similar semantics to the LLM-generated one in the test set.

Overall, I believe that the paper is nice and explores a potential direction that could be valuable in some use-cases, but the flaws and limitations previously mentioned make me question the practical efficiency of such a detector deployed at scale in the real world. I believe the paper could be much stronger if:

a) The authors would carefully build a large and semantically diverse database of generated texts and would add human-generated texts to the test dataset. The detection method would be significantly more convincing if it did not fail in this scenario.

b) The authors would design some similarity metric that would result in a large score only if the candidate text is some paraphrase of the target. This could potentially greatly reduce the number of false positives.

[1]: https://arxiv.org/pdf/1301.3781.pdf

**Questions:**

- I would like to see what happens if text generated by an LLM is kept in the database and some semantically similar entries generated by humans would be present in the test dataset.

- Following the first point, what would happen if you used DIPPER to paraphrase the human-generated text? Would there be higher similarity scores for the paraphrased samples if you have some semantically similar generated text in the database?

- Have the authors tried any similarity metric tailored for paraphrasing? Such a contribution would make the paper significantly stronger, in my opinion.

**Limitations:**

- The authors have extensively discussed potential limitations in the main paper and the Supplementary materials for both their DIPPER model and the retrieval-based detection systems. While the limitations regarding the computation necessary for a large-scale retrieval system can be somewhat alleviated, as discussed in section B.2 of the Supplementary, I believe the similarity-based scoring approach to be a significant limiting factor.

---

> ### Author Rebuttal · Authors · 2023-08-06
>
> We thank the reviewer for their thoughtful feedback! We are grateful that the reviewer appreciated our writing, paraphraser, the paraphrasing attacks on detectors, and discussion on the limitations of retrieval.
>
> The reviewer voiced concerns about the practicality of retrieval as a detection algorithm and mentioned technical challenges in implementing it at scale. We address these concerns below.
>
> >similarity score could also be large if the two texts have similar semantics...if one used encoder like word2vec…paper would be stronger if...authors design some similarity metric that would result in a large score only if candidate text is some paraphrase
>
> To the best of our understanding, the reviewer is referring to topical similarity between two texts, which could lead to high cosine similarity between non-paraphrases. This may be true for methods like word2vec, but our paper uses the P-SP embedding method which has been **explicitly trained for paraphrastic similarity using parallel data**. P-SP [4] achieves state-of-the-art performance on the STS benchmarks [1], which include plagiarism detection, measuring paraphrastic similarity on varying degrees of semantic overlap. To further illustrate the robustness of P-SP to non-paraphrases sharing topics, we conducted an experiment on the Par3 dataset. We found that the average P-SP score of actual human paraphrase pairs in Par3 is 0.76. In contrast, the P-SP of random pairs of paragraphs from the same book is just 0.09 (topically similar but not paraphrases).
>
> We have also added a discussion on “semantic collisions” in the “Global Rebuttal”. Overall, the chance of collisions between topically similar non-paraphrases is low because
> * the likelihood of semantic divergences between pairs exponentially increases with length
> * the most effective retrievers do not solely rely on semantics
> * unperturbed text will always have a 100% detection rate.
>
> >The paper would be stronger if authors would add human-generated texts to test set…with similar semantics to LLM-generated data.
>
> We have effectively done this in Sec. 5.2b, 5.3. The database in 5.2b contains three generations for the same prompt (hence strong topical overlap), and the test set includes the human-written output for the *same prompts*. Similarly, in 5.3, we have hundreds of database entries from the same PG19 book which closely share topical content. Our test set includes the human-written continuation for those prompts.
>
> >the authors have used human-generated continuations only to adjust the detection thresholds to maintain 1% FPR.. better methodology would be using parts of the human-generated data to fix a low FPR, and keep rest in test set.
>
> We do not think using human text for threshold adjustment removes them from the test set. A high FPR will push up the threshold, which in-turn will lower the true positive rate. This is equivalent to taking the y-coordinate for x=1% in a classifier’s ROC plot (TPR vs FPR). Moreover, in Fig 8 (Appendix), we plot the full ROC plots which consider every possible detection threshold. We find retrieval to be vastly superior and robust to paraphrasing compared to other methods on different thresholds.
>
> While we could use a random fraction of human-written text for threshold adjustment and evaluate the other fraction’s FPR, we expect it to result in a 1% +/- delta FPR since it’s in-distribution data. To validate this hypothesis, we empirically evaluated it in one setup (PG19, BM25), using 50% of the human text to estimate the threshold. We found the FPR on the other 50% of human text to be 0.8-1.2% across runs, and we expect variance to reduce with a bigger human dataset.
>
> However, we acknowledge that our paper does not test the out-of-distribution generalization of threshold adjustment (do thresholds for human dataset 1 give low FPR on human dataset 2?). We will do this in the next version.
>
> >Maintaining such a big database of generated text would be expensive from a storage perspective and computationally expensive
>
> In the “Global Rebuttal”, we provide a detailed analysis estimating these requirements for a ChatGPT-scale database. In summary, we estimate them to be relatively small. We estimate the ChatGPT database needs 5TB of storage space per month, and require just 100 seconds per retrieval on a CPU-only Macbook Pro. This is trivial compared to the scale at which Google Search (100,000+ TB index) and ChatGPT (10-15 seconds on a powerful 8x A100 GPU server) are currently operating. Moreover, major LLM providers (like OpenAI, Google) already have the infrastructure to host services like Google Search and ChatGPT at scale.
>
> >One needs access to all of the text generated by the model…limiting since the data would be available only to the entity that is maintaining the LLM API
>
> We agree that our detector can only be implemented by the LLM API provider, and have acknowledged this in Appendix B.1. However, major LLM providers may be incentivized to implement this detector, since it can help claim innocence in potential lawsuits about the origin of malicious AI-generated content. Moreover, there are increasing government discussions on regulating AI-generated content, and both the US and EU government have recommended major LLM providers to make their AI generations detectable [2, 3].
>
> >the method would only be viable for closed, proprietary models, and would not apply to open-source models
>
> This is a valid concern, and we address it in the “Global Rebuttal”. Overall, we agree with the reviewer that our detector is restricted to closed-source LLMs. However, most major LLM providers are hosting their LLMs behind closed APIs. Also, watermarking, the most promising alternative to retrieval, suffers from the same limitation. Other detectors either perform very poorly or are brittle against paraphrases.
>
> [1] https://aclanthology.org/S16-1081
> [2] https://tinyurl.com/usgovt-ai
> [3] https://tinyurl.com/eugovt-ai
> [4] https://aclanthology.org/2022.emnlp-demos.38

---

> > ### Comment · Reviewer_44uy · 2023-08-15
> >
> > I appreciate the authors' efforts in addressing the concerns raised by me and the other reviewers in the rebuttal.
> >
> > Upon reviewing the clarifications, I believe that this work can serve as a solid foundation in the efforts against paraphrasing attacks and can be a good contribution to NeurIPS. Consequently, I've adjusted my rating from 4 to 6.
> >
> > Nonetheless, I maintain that future iterations of this research should encompass more comprehensive experiments and a detailed examination of the datasets employed.
> >
> > I also suggest that for the camera-ready version of the paper, the authors should clarify the details regarding the human-generated test data in Sec. 5.2b and 5.3. As it stands, the test data composition remains somewhat ambiguous to me.

---

### Official Review · Reviewer_k3DJ · 2023-07-04

**Soundness:** 3 good
**Presentation:** 3 good
**Contribution:** 3 good
**Rating:** 6
**Confidence:** 3

**Summary:**

The authors developed a powerful paraphrase generation model called DIPPER to test the robustness of AI text detection algorithms. DIPPER successfully evaded several detectors by paraphrasing text generated by large language models. To improve detection, they proposed a defense mechanism based on retrieving similar generations from a database, which detected 80% to 97% of paraphrased text while misclassifying only 1% of human-written sequences as AI-generated.






**Strengths:**

1. Testing the robustness of existing detectors of AI-generated text is very interesting and important.

2. A defence method is proposed to handle the paraphrasing problem discovered in this paper.

3. The experimental results are solid and promising.

**Weaknesses:**

1. The proposed method needs the store of all AI-generated texts. I wonder whether it is practical for popular LLMs which many users and queries, like ChatGPT.

2. More detection methods should be incorporated in experiments.

3. It would be better if more datasets and tasks are used.

**Questions:**

Is it practical for popular LLMs which many users and queries, like ChatGPT to store all their generated texts?

**Limitations:**

The authors have addressed the limitations.

---

> ### Author Rebuttal · Authors · 2023-08-06
>
> We thank the reviewer for their useful feedback and support for our paper! In particular, we appreciate the reviewer for highlighting that 1) we study important research questions on robustness of AI-generated text detection; 2) our defense mechanisms for our attacks; 3) our paper shows solid experimental results.
>
> The reviewer voiced some concerns about storage requirements, and requested experiments on more detection methods and datasets. We address these below.
>
> > The proposed method needs the store of all AI-generated texts. I wonder whether it is practical for popular LLMs which many users and queries, like ChatGPT.
> > Is it practical for popular LLMs which many users and queries, like ChatGPT to store all their generated texts?
>
> We think it is very much practical at ChatGPT’s current usage rate (1.5-2B monthly visits) to store AI-generated outputs. We estimate it will take about 5TB storage space per month (detailed calculations in the “Global Rebuttal”), similar to the size of a personal portable hard-disk. Moreover, ChatGPT and Google Bard already store user’s chat history for improving their models with RLHF [1, 2]. Moreover, the major LLM providers like Google and OpenAI already have the storage infrastructure in-place which they use to host services like Google Search (100,000TB+ index size) and ChatGPT / GPT3.5. In our “Global Rebuttal” we also extensively discuss scalability in terms of compute and accuracy.
>
> > More detection methods should be incorporated in experiments.
> > It would be better if more datasets and tasks are used.
>
> Our experiments already cover a comprehensive set of five detection algorithms (DetectGPT, watermarking, OpenAI classifier, GPTZero and RankGen). At that time our research was conducted, we included every state-of-the-art contemporary AI-generated text detector we could find. Moreover, our experiments cover two practically relevant tasks modern large language models are used for: long-form question answering (six domains) and open-ended generation (two domains). Finally, our experiments are performed on outputs from three large language models (GPT2-XL, OPT-13B, GPT3.5 davinci-003), to provide a wide diversity of LM sizes and properties.
>
> Nevertheless, we acknowledge that our scaled retrieval experiments (Section 5.3) cover outputs from only one kind of language model in two domains. As promised in the “Global Rebuttal”, we will conduct similar experiments on other AI-generated text databases like GPT4All and ShareGPT, which while smaller in size, are more diverse in nature than the RankGen training data.
>
> [1] - https://openai.com/blog/new-ways-to-manage-your-data-in-chatgpt
> [2] - https://support.google.com/bard/answer/13594961

---

> > ### Comment · Reviewer_k3DJ · 2023-08-18
> > **Thanks for the response**
> >
> > Thank the authors very much for the detailed response. It well addressed my questions mentioned in my review.

---

### Official Review · Reviewer_VDA7 · 2023-07-05

**Soundness:** 3 good
**Presentation:** 4 excellent
**Contribution:** 3 good
**Rating:** 8
**Confidence:** 4

**Summary:**

This paper investigates the robustness of AI-generated text detection algorithms to paraphrasing. The authors train a language model to paraphrase text in an attempt to evade detection algorithms. The proposed model leverages longer contexts than existing sentence-level paraphrasers and offers users control over content diversity and reordering. After demonstrating the vulnerability of current detection algorithms to the paraphrasing model, the authors propose a retrieval-based defense in which the outputs of a language model are stored and matched against possible AI-generated text. The authors find their retrieval-based defense achieves superior robustness against paraphrasing when compared to existing watermarking and classification defenses.

**Strengths:**

The training scheme for DIPPER is clever and well motivated (e.g. control over content re-ordering and diversity). I think the paper benefits from including both an improved attack on AI text-detection and a simple but interesting defense.

Very thorough experiments.

I appreciate the discussions of the limitations of retrieval-based defenses in the appendix.

**Weaknesses:**

The authors do not appear to evaluate any non-DIPPER paraphrasers in their experiments. The claim in line 232 that "our ablations in Appendix C show that these paraphrasers have lower quality and are less compatible with the prompt as DIPPER paraphrasers" seems misleading, as DIPPER ablations are not the same as evaluations of existing off-the-shelf paraphraser models. While the paper presents strong results for DIPPER, I think the authors should either re-word this claim to be more clear or provide direct experimental comparisons to other paraphrasers.

Paraphrasing attacks might not be considered effective if they degrade the original text in certain ways (e.g. introduce grammatical errors, change the "tone" or "voice" of the text) even if they preserve some notion of semantic similarity. The results in the main paper heavily emphasize the semantic similarity metric of Wieting et al. as a measure of paraphrased text quality, although paraphrasers are evaluated under other metrics (e.g. human evaluations, perplexity) in the appendix. As things stand, it is not immediately clear in the main body of the paper whether DIPPER strongly preserves the quality of the original text beyond some notion of semantic similarity. This should probably be clarified by summarizing the results of the additional quality evaluations from Appendix C.

**Questions:**

If the underlying language model is itself capable of diverse generations, is it possible that we might see a "saturation" problem in certain topic areas? E.g. if hundreds of students use the language model to generate one-paragraph summaries of the Gettysburg address, all of which are stored in the database, will this provide a kind of "semantic coverage" of the topic such that even human-generated summaries of the Gettysburg address would be flagged as AI-generated?

I think the authors' proposed retrieval defense is promising, and their proposed improvements in Appendix B.2 are interesting, but I'm not sure whether the paper effectively argues for the feasibility of retrieval-based defenses at scale. In Appendix B.1, line 680, the authors state that "At a conservative rate of 5M queries a day, [an AI text generation] database will have almost two billion entries in a year." While the authors perform experiments to validate their retrieval-based defense at a scale of 15 million entries, it is not immediately clear how this performance would extrapolate to a database three or more orders of magnitude larger (as in their hypothetical). Along these lines, it would be helpful if the authors could point to one or more analogous text-retrieval systems capable of operating at the aforementioned hypothetical scale with the precision required for AI text detection.

**Limitations:**

Yes

---

> ### Author Rebuttal · Authors · 2023-08-06
>
> We thank the reviewer for their thoughtful feedback and support for our paper! In particular, we are grateful for reviewer’s appreciation for our 1) novel paraphraser training algorithm DIPPER; 2) attacks on AI-generated text detectors using DIPPER; 3) novel defense mechanism and a discussion of its limitations; 4) thorough experiments in all these fronts.
>
> The reviewer voiced a few concerns about our paraphraser evaluations, and had a few questions about the scalability of our retrieval-based defense mechanism. We address them below.
>
> > claim in line 232 that "our ablations in Appendix C…" seems misleading, as DIPPER ablations are not the same as evaluations of existing off-the-shelf paraphraser models.
>
> We agree with this concern and will reword L232 to better reflect our contribution. In particular, our stance is that we expect non-DIPPER paraphrasers to also evade detection. However, we expect DIPPER will produce higher quality paraphrases than a non-contextual sentence-level alternative (as shown in our Appendix). Another advantage of using DIPPER is its fine-grained control knobs to vary diversity to find a sweet spot of retaining semantics, while fooling detection systems.  We did not find these traits in other off-the-shelf paraphrasers, as discussed in Appendix D.1. We will move some text from the Appendix to the main body to further highlight this.
>
> > Paraphrasing attacks might not be considered effective if they degrade the original text in certain ways (tone, style).. As things stand, it is not immediately clear in the main body of the paper whether DIPPER strongly preserves the quality of the original text.. This should probably be clarified by summarizing the results of the additional quality evaluations from Appendix C
>
> As the reviewer pointed out, we have included several additional automatic and human paraphrasing evaluations in Appendix C to showcase the strengths of DIPPER. We will summarize these results in the main body of the paper. We agree with the reviewer that paraphrases are likely to modify style as shown in [1]. However, as mentioned in the previous paragraph, an important property of DIPPER is to provide fine-grained diversity control. Our lexical and order diversity control knobs allow an attacker to modify a generation *just enough* to evade detection: lower the value of the knobs, lesser the stylistic modification.
>
> > If the underlying language model is itself capable of diverse generations, is it possible that we might see a "saturation" problem in certain topic areas (the Gettysburg Address)?
>
> This is a great point about semantic overlap within the database on popular topics. We address this in detail in the “Global Rebuttal” under “semantic collisions”. In summary, we believe that the chance for semantic collisions is low because:
> * retrieval-based detection uses pairwise comparisons, and the likelihood of semantic divergences between pairs exponentially increases with length;
> * the most effective retrievers do not solely rely on semantics;
> * unperturbed text will always have a 100% detection rate.
>
> > I'm not sure whether the paper effectively argues for the feasibility of retrieval-based defenses at scale.
>
> We extensively discuss this in the “Global Rebuttal” to all reviewers, on axes of storage, compute and accuracy. Overall, we are quite optimistic about the feasibility of retrieval in terms of storage requirements (5TB for a database with 1.5-2B generations) and compute requirements (just 100 seconds for retrieval against a 1.5B database on a Macbook Pro). Importantly, the major players providing LLM API services (Google, OpenAI, Microsoft) have vastly superior computational infrastructure already in place to power services like Google Search, ChatGPT and Bing at scale.
>
> In terms of accuracy, we are optimistic looking at our scaling curves (Figure 5a shows just 0.8% drop from 1M to 10M in BM25). Our experiments already use the largest publicly available AI-generated dataset (to the best of knowledge), and collecting a billion-scale dataset with ChatGPT will cost $1M.
>
> > it would be helpful if the authors could point to one or more analogous text-retrieval systems capable of operating at the aforementioned hypothetical scale
>
> Traditional information retrieval has a slightly different setup compared to our retrieval-based detection. Queries tend to be more information seeking (rather than looking for exact matches or paraphrases), and recall@k is also important besides precision. Nevertheless,  we think Google Search is an excellent example of text-retrieval operating at scale, since Google’s search index is over 100B webpages [3]. In academic literature, we found a few examples of experimental setups operating at a billion-scale [4, 5]. [4] shows a precision@1 of 60% for time-aware retrieval.
>
> [1] https://arxiv.org/abs/2010.05700
> [2] https://eval.ai/web/challenges/challenge-page/1897/leaderboard/4475
> [3] https://www.google.com/search/howsearchworks/how-search-works/organizing-information
> [4] https://sobre.arquivo.pt/wp-content/uploads/creating-a-billion-scale-searchable-web-archive.pdf
> [5] https://arxiv.org/abs/2110.06125

---

> > ### Comment · Reviewer_VDA7 · 2023-08-16
> > **Response to authors**
> >
> > I thank the authors for their detailed reply to my review and for their general rebuttal. I think the authors have adequately addressed my concerns regarding the technical challenges of a retrieval-based defense, and I agree that the scale of the retrieval experiments performed is reasonable given the presumptive cost of creating a novel LLM text database. I think the paper will be significantly stronger with the authors' proposed modifications to address reviewer critiques (many of which were shared by multiple reviewers, and thus would likely be raised by readers). I have adjusted my score accordingly.

---

### Official Review · Reviewer_KFgu · 2023-07-06

**Soundness:** 4 excellent
**Presentation:** 4 excellent
**Contribution:** 3 good
**Rating:** 8
**Confidence:** 5

**Summary:**

The submission "Paraphrasing evades detectors of AI-generated text, but retrieval is an effective defense" investigates detection methods for   text output generated by modern large language models. The contribution of this submission consist of two parts. First, the submission describes, trains and provides a state-of-the-art language paraphrasing model. This paraphrasing model is then used to critically evaluate a number of detection schemes for their robustness against paraphrasing, finding that many modern detection approaches are not robust to paraphrasing attacks that leave the semantics of the original text unchanged, but modify its wording.

In a second part, the authors use this finding as motivation to describe a detection strategy based on semantic retrieval.

**Strengths:**

This is a great submission. It that starts out with a clear hypothesis about the detection of LLM-generated text and executes its two mechanical parts - First generating the paraphraser, and then constructing the retriever, very well.

In more detail, the construction of a paraphrasing data through re-alignment of paragraph-level parallel translations is novel to me and quite interesting. The authors make the reasonable choice of finetuning existing encoder-decoder models for paraphrasing based on this corpus, a choice that they experimentally validate to work well.

Then, they rightfully point out that the detection of generated can be cast as a retrieval problem, which they show indeed simplifies the problem, and allows for the leverage of existing similarity search tools. Just in case, I explicitely want to point out, that I don't consider the use of existing similarity search tools a weakness, but instead a correct conclusion based on the insight of the authors that the problem can understood in this manner. In a detailed evaluation, the authors compare LLM detection based on retrieval with other approaches, finding this to be a robust choice.


Finally, I especially want to highlight the immediate practical value that the paraphrasing model would have as a tool for the community. Previous papers have often only hinted at the possibility of paraphrasing, or used general-purpose APIs to attempt to provide accurate paraphrases, and this has limited evaluations of the very practical threat model of paraphrasing in the literature. With the release of this paper and the paraphrasing model provided here, the authors would provide a practical tool to the community that will suddenly makes investigation of this threat model much more feasible.

**Weaknesses:**

I only see a few minor weaknesses, which I will point out below. These mainly orbit around questions of "why", which this submission does not always contain.

* Why would these detectors break under paraphrasing? While we do observe that the detectors based on outlier detection and classifier methods break after paraphrasing, aren't some of these detectors kind of correct in their assessment that the text is not written by the original model anymore? This does not apply to all detectors, but those that claim to only detect a specific model (or model family) are not entirely incorrect? To phrase this question differently, would paraphrasing also break detectors that classify "generic" machine text, and are not specialized to detect particular models or model families?


* Why is FPR low for detection?There seems currently no way to estimate the reliability of the retrieval-based detector formally and to guarantee a certain FP rate to my understanding? Is this a principal limitation, or could the method be modified to use a threshold not calibrated from existing data? Ideally, in a way that includes the size of the corpus, allowing for estimates of performance of this approach at larger scales than can be tested empirically in this work?

* Is there a corpus size at which retrieval stops being meaningful, if it is based only on semantics? I could imagine, for the sake of the argument, that many school essays that summarize existing arguments about some fact or historical event on per-paragraph level, might be semantically the same. These would also be semantically similar to some output when the model was at one point queried with this topic separately?


In any case, the submission stands strong based on the empirical evidence it provides, and some of these questions might be left to future work.

**Questions:**

A few smaller questions:

* What about random substring detection? To my understanding, the submission only tests queries where the back-end of the query is removed. Would a detection of a random substring from the query perfom equally well? Bonus question: What about retrieval for several substrings from different corpus entries, i.e. what if a document is constructed based on text generated by mixing and matching multiple queries to the language model API?

* How large is the Par3 dataset? Could the authors briefly comment on dataset size and finetuning compute required for the paraphraser in the main body?

* I found the different FPR for DetectGPT in  a part of Table 1 somewhat confusing. I see that the authors want to be polite here, but to me it would be clearer to indicate that this detection method really scores 0% at the given FPR. A 20% table could be included in the appendix, although I agree with the authors that there is no practical value to detection schemes with this FPR.

**Limitations:**

There are a number of ethical implications to storing all user-generated text on the server for retrieval, concerning data privacy. It would be great if the authors could briefly  comment on this also in the main body, as this question is currently only discussed in Appendix B.1.4.

One question I have, related to this discussion of ethics, (this is a bit of an aside, the paper is great without answering this), is how retrieval would interact with regulations that include the "right-to-be-forgotten", like GDPR. Could a user ask the company to delete their generations, to prevent detection? Or should it be argued that detection is "sufficient cause" to stop deletion of user data?


I also think there are soom possible solutions to this question, where reconstruction of x_i from y_i can be ruled out, possibly via bloom filters, or (minimal) differential privacy?

---

> ### Author Rebuttal · Authors · 2023-08-06
>
> We thank the reviewer for their detailed and thoughtful feedback, and for strongly supporting our paper! In particular, we are grateful to the reviewer for supporting our 1) novel paraphraser and its open-sourcing; 2) our experimental analysis testing the robustness of AI-generated text detectors; 3) our casting of AI-generated text detection as a retrieval problem and its downstream robustness.
>
> The reviewer further asks a number of interesting questions related to paraphrasing as an attack and retrieval as a defense. We address them below.
>
> > would paraphrasing also break detectors that classify "generic" machine text, and are not specialized to detect particular models
>
> Our experiments do show that paraphrasing drops performance in all model-specific detectors we tried (watermarking, DetectGPT) and two out of three model-agnostic detectors (OpenAI, GPTZero). However, it’s hard to fairly compare the relative effect of paraphrasing on these two classes due to a large performance gap between them in the first place. In Table 1, we see that even without paraphrasing, model-specific detectors significantly outperform model-agnostic detectors. OpenAI has even gone so far to remove their model-agnostic classifier from their website due to low accuracy [1]. Overall, we observe that model-specific watermarking is the most robust to paraphrasing attacks despite the large drop. It’s also important to note that paraphrased text is more “AI-generated” in some sense. One of our model-agnostic detectors (RankGen) thinks it is indeed so (7% vs 1% TPR for GPT3.5), but its low overall TPR makes it unusable as a detector.
>
> It is technically true that model-specific detectors are not designed to detect outputs not entirely generated by the model itself. However, given their strong performance over model-agnostic methods, and the large risk of perturbation attacks (human-edited or automatically-edited), we think it’s important for model-specific detectors to bake perturbation robustness in their design for better downstream usability.
>
> > There seems currently no way to estimate the reliability of the retrieval-based detector formally and to guarantee a certain FP rate to my understanding?
>
> Similar to DetectGPT / classifiers, we believe the threshold needs to be estimated empirically on the underlying data distribution. We will add some analysis on this in the next version, and analyze the out-of-distribution robustness of thresholds chosen on a subset of human data. A formal relationship between FPR and the threshold (like in watermarking) may be possible using information about the density of the retrieval database in the semantic vector space. We leave this exploration for future work.
>
> > Is there a corpus size at which retrieval stops being meaningful, if it is based only on semantics?…school essays that summarize existing arguments about some fact or historical event…might be semantically the same.
>
> This is a great point about semantic overlap within the database on popular topics. We address this in the “Global Rebuttal” under “semantic collisions”. In summary, we believe that the chance for semantic collisions is low because,
> * retrieval-based detection uses pairwise comparisons, and the likelihood of semantic divergences between pairs exponentially increases with length
> * the most effective retrievers do not solely rely on semantics
> * unperturbed text will always have a 100% detection rate.
>
> > What about random substring detection? What about retrieval for several substrings from different corpus entries
>
> Our experiments in Fig 5b experimented with truncated paraphrases as queries. Below, we present results for other query types that the reviewer suggested. We adopt the same setup as 5b: PG19-BM25, 1% FPR:
>
> Results in paper:
> full length unperturbed query: 100%
> full length DIPPER query: 98.2%
> 50% trunc DIPPER query: 72.6%
>
> New results:
> 50% random unperturbed substring: 86.2%
> 50% of two different unperturbed queries concat: 94.7%
> 50% random DIPPER substring: 68.8%
> 50% of two different DIPPER queries concat: 56.1%
>
> Overall, we see that unperturbed random substrings (from single or multiple generations) can still be detected quite easily. However, adding DIPPER paraphrasing on top of that reduces accuracy.
>
> > How large is the Par3 dataset?… fine tuning compute?
>
> Our processed Par3 dataset has 6.3M pairs (which we will open source). We noticed convergence on held-out novels within 12 hours on 64 cloud TPUv3 chips.
>
> > … how retrieval would interact with regulations that include the "right-to-be-forgotten", like GDPR. Could a user ask the company to delete their generations, to prevent detection? Or should it be argued that detection is sufficient cause to stop deletion?
>
> We agree with the reviewer that AI-generated text detection is a sufficient cause to temporarily override deletion requests [2]. We believe that AI-generated text detection could fall under the following GDPR guidelines for overrides: (1) “freedom of expression and information”,(2) “establishment of a legal defense or in the exercise of other legal claims”, and possibly (3) “comply with a legal ruling or obligation” in the future. As an example of this, while OpenAI allows users to delete their chat history [3], they retain them for 30 days and can review it if required to monitor for abuse. We also agree with the reviewer that differential privacy or scrubbing sensitive attributes can mitigate the privacy issue to some extent. We will mention this in the next version, and also move our discussion on data privacy from Appendix B.1.4 to the main body as requested by the reviewer.
>
> > I found the different FPR for DetectGPT in Table 1 confusing
>
> We agree with this concern about our presentation. We will clarify this in the next version.
>
> [1] https://openai.com/blog/new-ai-classifier-for-indicating-ai-written-text
> [2] https://gdpr.eu/right-to-be-forgotten
> [3] https://openai.com/blog/new-ways-to-manage-your-data-in-chatgpt

---

> > ### Comment · Reviewer_KFgu · 2023-08-12
> > **Thanks**
> >
> > Thank you for the additional clarification and experiments, I have no further questions!

---

### Official Review · Reviewer_qaDF · 2023-07-07

**Soundness:** 3 good
**Presentation:** 2 fair
**Contribution:** 3 good
**Rating:** 6
**Confidence:** 4

**Summary:**

This paper first demonstrates the vulnerability of existing
AI-generated text detectors to paraphrases, and then proposes a
retrieval-based method to alleviate the issue.  For the first
experiment, the authors trained a paragraph-level paraphrase
generation system, called DIPPER, by fine-tuning an existing
text-to-text model, i.e., T5-XXL.  To this end, the authors compiled a
dataset comprising pairs of paragraphs with two indication parameters
to control lexical diversity and content reordering.  The proposed
retrieval-based detector assumes the LLM API providers to save all
their generated contents and to offer an interface for access the
database.  The authors experimented with a corpus of 15 million such
examples, and shows that this approach is more robust to paraphrased
texts compared to existing detectors.


**Strengths:**

- A paragraph-level paraphraser with the recipe to build it.

- Demonstration of the vulnerability of existing AI-generated text
  detectors to paraphrases.

- A simple but effective retrieval-based method for robust
  AI-generated text detection.


**Weaknesses:**

- The underlying assumption of the proposed method is not likely to be
  realistic.  We can call LLM API providers for their ethics, but
  there is no means to enforce them to conform to this approach.
  Beside the providers, nowadays there are also increasing number of
  publicly available strong LLMs, and we cannot regulate personal uses
  of LLMs and AI-generated texts.  I acknowledge the discussion in
  Appendix B.1, but nothing is discussed in the main paper.
  [Through rebuttal, the authors explained some obligations of LLM
  providers and promised to include its summary in the main paper.]

- Scalability of the proposed approach is not sufficiently evaluated.
  ll.297-298 states "a popular LLM API may serve millions of queries a
  day" and it is likely to be true, given ChatGPT acquired more than
  100 million users in the first two months and more than 13 million
  unique users a day.  Compared to this, the experiment in this paper
  uses up to only 15 million AI-generated texts, it is unclear whether
  the proposed method is feasible (memory and time in addition to
  accuracy) with billions or trillions of such text.  I acknowledge
  the discussion in Appendix B.2, but nothing is discussed in the main
  paper.
  [The response reports on 100 seconds per retrieval at the client side
  but this is misleading since the data store and retrieval operations
  must locate on the server (LLM provider) side.  However, I
  understand that the speed would not matter given the strong facility
  at LLM providers and high parallelizability of the retrieval task.
  The experiment with 15 million instance sounds not realistic but I
  understand that we should accepted this if the authors are not in
  giant tech.]


**Questions:**

- In l.127, $L$ is defined as "unigram token overlap" but which of $p$
  and $q$ is used to determine the denominator? [harmonic mean]

- Input sequences in l.132 and l.180 contains ordinary tokens:
  "lexical", "order", "=", and ",".  Did the authors use them without
  escaping as "<p>" and "</p>" ?  If so, wouldn't it distort the
  embeddings for these tokens? [not escaped, following convention.]

- How many training instances were used for obtaining DIPPER?
  [6.3 million pairs, which should be included in the main paper.]

- I'd like to suggest to tidy the layout.  For instance, Tables 1, 2
  and Figure 5 appear in the previous page of their first mention in
  the main text; Figure 3 shows a table; the mention of Figure 6
  appears before Figures 4 and 5; Figure 6 is embedded in an
  irrelevant paragraph. [the authors promised to do so.]

- l.264: $L$ has already been used in l.127.


**Limitations:**

Yes. The authors describe the limitations in Appendix B and ethical considerations in Appendix A.

---

> ### Author Rebuttal · Authors · 2023-08-06
>
> We thank the reviewer for their detailed and thoughtful feedback! We thank the reviewer for supporting our contributions on paragraph-level paraphrasing, attacking AI-generated text detectors, and introducing a retrieval-based defense mechanism.
>
> The reviewer voiced concerns about the practicality of the proposed retrieval-based detection algorithm, some of which the reviewer noted were discussed in our Appendix (and which we will summarize in the main body of the next version). While we address these concerns below, we want to emphasize that retrieval-based detection is only one of our three contributions. Besides retrieval, our contributions include: 1) a novel paragraph-level paraphraser which we will open-source (data + models) for the community to use; 2) a thorough experimental study attacking five AI-generated detectors on outputs from three diverse large language models on two real-world tasks. We will fully open-source the code and datasets for reproducibility.
>
> >  We can call LLM API providers for their ethics, but there is no means to enforce them to conform to this approach (retrieval-based detection)
>
> We believe that LLM API providers may actually be enforced or even incentivized to store their model-generated outputs:
>
> * There is a substantial push from both the US and European governments to regulate companies to make their AI-generated text/images detectable. For instance, two weeks ago, Google, OpenAI and Meta made voluntary commitments to watermark their AI-generated content [1]. Similarly, the European government recently pushed for rulings to make AI-generated content detectable [2].
>
> * Companies are already storing their model-generated outputs. For instance, both ChatGPT and Bard are storing chat histories to help improve their products with RLHF training. Bard stores chat history for 18 months by default [6], and OpenAI stores the history for 1 month even if users choose to opt-out [5].
>
> * In a hypothetical scenario where there is a lawsuit about the origin of some malicious AI-generated content, maintaining a database of previously generated responses could be a reliable method to prove innocence, given its strong performance over competing AI-generated text detectors.
>
> > it is unclear whether the proposed method is feasible (memory and time in addition to accuracy) with billions or trillions of such text
>
> In the “Global Rebuttal”, we provide a detailed analysis estimating these requirements for a ChatGPT-scale database. In summary, we estimate them to be relatively small. Specifically, we estimate that a month’s database of ChatGPT outputs needs 5TB of storage space per month and requires just 100 seconds per retrieval on a CPU-only Macbook Pro. This is trivial compared to the scale at which Google Search (100,000+ TB index) and ChatGPT (10-15 seconds on a powerful 8x A100 GPU server) are currently operating. Moreover, major LLM API providers (like OpenAI, Google) already have the infrastructure in-place to support services like Google Search and ChatGPT at scale.
>
> In terms of accuracy, we are optimistic looking at our scaling curves (Figure 5a shows just 0.8% drop from 1M to 10M in BM25). Our experiments already use the largest publicly available AI-generated dataset (to the best of knowledge). Collecting a billion-scale dataset with ChatGPT will cost $1M, making the experiment only possible using OpenAI or Google’s private databases.
>
> > nowadays there are also an increasing number of publicly available strong LLMs, and we cannot regulate personal uses of LLMs and AI-generated texts
>
> This is a valid concern, and we address this in the “Global Rebuttal”. Overall, we agree with the reviewer that our detector is restricted to closed-source LLMs. However, most major LLM providers are hosting their LLMs behind closed-source APIs. Additionally, watermarking, the most promising alternative to retrieval, also suffers from the same limitation and cannot be done on open LLMs. Other detectors either perform very poorly or are brittle against paraphrasing attacks.
>
> > I acknowledge the discussion in Appendix B.1, but nothing is discussed in the main paper…. I acknowledge the discussion in Appendix B.2, but nothing is discussed in the main paper.
>
> We will add a summary of limitations of retrieval-based detection to the main body of the paper.
>
> > In L127, L is defined as "unigram token overlap" but which of p/q and is used to determine the denominator?
>
> In L127 we use an F1 score to determine unigram token overlap between p and q. So this considers both p and q in the denominator before the harmonic mean.
>
> > L132, L180 contain ordinary tokens: "lexical", "order", "=", and ",". Did the authors use them without escaping as "<p>" and "</p>"?
>
> We did not escape “lexical” / “order” in order to leverage the pre-trained embeddings for these words. We don’t think this should have any major effect on training, because we are using pre-trained 11B-sized deep transformers models which are extremely compositional in nature. While embeddings may get slightly distorted, we expect higher layers to compose information from consecutive tokens to infer the differences between control tokens and content tokens. We also note that unescaped control tokens are standard practice in T5 fine-tuning (Fig 1 in [1]).
>
> > How many training instances were used for obtaining DIPPER?
>
> We used a dataset of 6.3M pairs (which we will open source). We noticed convergence on held-out novels within 12 hours on 64 cloud TPUv3 chips.
>
> > I'd like to suggest to tidy the layout
>
> We agree with these presentation issues, and will clean them up in the next version.
>
> [1] https://tinyurl.com/usgovt-ai
> [2] https://tinyurl.com/eugovt-ai
> [3] https://arxiv.org/abs/1910.10683
> [4] https://openai.com/blog/new-ways-to-manage-your-data-in-chatgpt
> [5] https://support.google.com/bard/answer/13594961
> [6] https://openai.com/blog/new-ai-classifier-for-indicating-ai-written-text

---

> > ### Comment · Reviewer_qaDF · 2023-08-12
> > **Read the responses**
> >
> > Thank you for your response. I'll update my review accordingly.

---

### Author Rebuttal · Authors · 2023-08-06

We are very grateful to the reviewers for their detailed feedback. While we address each reviewer’s questions in the individual rebuttals, we use this “global rebuttal” to address concerns shared by multiple reviewers.

We thank the reviewers for supporting the three contributions in our paper:
* a novel discourse paraphrasing model and its open-sourcing
* a comprehensive set of effective paraphrasing attacks on modern AI-generated text detectors
* a novel defense using information retrieval and extensive discussion of its limitations.

Many of the concerns center around our third contribution of retrieval-based detection. These concerns can be categorized as:
* issues of scalability (storage, compute, accuracy)
* limitation to closed-source LLMs
* semantic collisions

Below, we address each in detail.

**Scalability of retrieval - storage**: We estimate ChatGPT’s outputs to take 5TB space monthly (similar to a personal portable hard-disk) via the following calculations. ChatGPT currently gets about 2B monthly visits [3]. Assuming an average response length of 500 tokens per session, this corresponds to 1 trillion tokens. Similar in size to LLaMA’s training data, this needs 5TB space [2]. However, 5TB is a small amount of storage compared to the industrial scale of information retrieval. For example, the Google Search index is over 100,000TB and has 100B+ pages [1]. Major LLM service providers (like Google, OpenAI) already have complex storage infrastructure to facilitate this defense. Additionally, it’s likely that they already store their model outputs for future RLHF purposes.

**Scalability of retrieval - compute**: Our retrieval experiments, conducted on a 14-core CPU (similar to a Macbook Pro), took 1 second per retrieval on a 15M sized corpus. Extrapolating to a corpus of ChatGPT’s monthly usage (100x) would need 100 seconds/retrieval on a Macbook. However, this is fully parallelizable, and can make use of GPUs (Google searches 100B+ entries in < 1 sec). Moreover, efficient similarity search has powerful libraries like FAISS available. For comparison, ChatGPT itself takes 10 seconds/response, possibly using a powerful 8-GPU A100 server [4]. Major LLM providers have massive compute clusters, and we believe the computational requirement of retrieval is much lower than hosting LLMs in the first place, which these providers are already adept at. Moreover, our proposed ideas in B.2 can further reduce compute costs.

**Scalability of retrieval - accuracy on larger databases**: Our experiments were conducted on the RankGen training set [7], which is the largest publicly available database of AI-generated text that we are aware of (15M generations each in four domains). Besides this, there are a few other datasets of AI-generated text such as GPT4All (809K) and ShareGPT (350K generations). While these datasets are much smaller than ours, we will experiment with them in the next version to add more diversity to our results. We note that it is extremely expensive and time consuming to create a corpus of AI-generated text from scratch: at a cost of `$`0.001 per 500-word response, collecting a billion ChatGPT outputs would cost $1M and take a long time to collect due to rate limits. Hence, a billion-scale experiment is likely only possible with Google/OpenAI’s private database.

Overall, we are optimistic about our scaling plots (Fig 5a), and see just a 0.8% drop moving from a 1M to 10M database (PG19-BM25). We emphasize that BM25 is a basic retriever, and is not optimized on our task. Information retrieval literature has many powerful retrievers, and we have suggested a dense retrieval mechanism in B.2 which can be optimized on the underlying retrieval corpus. Moreover, retrieval can easily be used in tandem with other detectors like watermarking. Finally we note that our paper is the first proof-of-concept that shows a retrieval-based detector could work, and we anticipate future work to build upon it.

**Retrieval is limited to closed-source LLMs**: We agree with this and will add it to our limitations. However, besides Meta, all major LLM providers (OpenAI, Google, Anthropic, Microsoft, Cohere) operate their LLMs behind closed APIs. It’s also important to note that watermarking, the most promising alternative to retrieval, also has this limitation. Since watermarks are added during decoding rather than into the model weights, users of open LLMs are free to generate text without watermarks. While other alternatives (DetectGPT, classifiers) don’t suffer from this issue, our paper shows that they either have low accuracy, or are extremely vulnerable to paraphrasing. In fact, OpenAI recently took down their classifier due to low accuracy [5].

**Semantic collisions**: Reviewers raised a concern about our retrieval database saturating with entries having similar semantics (especially for popular topics), which will harm detection at scale. In response, we note (and will update our paper to include) that:
* Like other detectors, retrieval works best on longer sequences (Fig 5b). Long generations exponentially increase the likelihood of semantic divergences between pairs of entries.
* Retrieval compares the input against the top-1 match, not top-k. For false-positive inputs on popular topics, the top-k entries *together* are more likely to cover input semantics (recall) rather than top-1 (precision).
* The most effective retrievers use a combination of neural semantic encoders and token overlap scores [6]. We also show this, BM25 beats P-SP at detection. BM25 is not fully semantic driven: it uses TF-IDF token overlap.
* The retrieval accuracy for unperturbed AI-generated text is always 100%, just like exact match searches in Google Search. Retrieval is also effective on substrings of unperturbed text (see rebuttal to KFgu).

[1] tinyurl.com/ggsdb
[2] tinyurl.com/fbllama
[3] tinyurl.com/chatnyp
[4] tinyurl.com/chatgpu
[5] tinyurl.com/oaicls
[6] tinyurl.com/beirev
[7] tinyurl.com/rrkgn

---

### Decision · Program_Chairs · 2023-09-21

**Decision:**

Accept (poster)

**Comment:**

This is a solid paper with three key contributions: a novel paragraph-level paraphrase model, an “experimental study attacking five AI-generated detectors", and a new detection/defence method based on semantic retrieval. In the words of one of the reviewers, this paper will "serve as a solid foundation in the efforts against paraphrasing attacks and can be a good contribution to NeurIPS", while the approach will have good practical value for the community. However, the authors should ensure they fully address all of the reviewers’ concerns in the camera ready at least as extensively as they have done here.